# Comprehensive Non-targeted Molecular Characterization of Organic Aerosols in the Amazon Rainforest

Denis Leppla*[1], Stefanie Hildmann*[1], Nora Zannoni[2,a], Leslie A. Kremper[2,c], Bruna A. Holanda[2,c], Jonathan Williams[2,3] Christopher Pöhlker[2], Stefan Wolff[2,b], Marta Sà[4], Maria Christina Solci[5], Ulrich Pöschl[2], Thorsten Hoffmann[1]

[1]Chemistry Department, Johannes Gutenberg - University, Mainz, 55128, Germany
[2] Multiphase Chemistry Department, Max Planck-Institute for Chemistry, Mainz, 55128, Germany
[3]Climate and Atmosphere Research Center, The Cyprus Institute, Nicosia, Cyprus
[4]Large-Scale Biosphere–Atmosphere Experiment in Amazonia (LBA), Instituto Nacional de Resquisas da Amazônia/INPA, Manaus, AM, Brazil
[5]Department of Chemistry Universidade Estadual de Londrina, Londrina, PR, Brazil
[a]now at: Institute of Atmospheric Sciences and Climate, National Research Council (CNR-ISAC), 40129 Bologna, Italy
[b]now at: German Weather Service, Offenbach am Main, 63067, Germany
[c]now at: Hessian Agency for Nature Conservation, Environment and Geology, Wiesbaden, Germany
*These authors contributed equally to this work.

*Correspondence to*: Thorsten Hoffmann (t.hoffmann@uni-mainz.de)

## Abstract

The Amazon rainforest plays a crucial role in the global climate system, hydrological cycle, and earth's energy balance. As one of the planet's least industrialized regions, it allows investigation of organic aerosol formation and constituents under almost pristine conditions. Nevertheless, human activities are known to affect this ecosystem – especially during the dry seasons. In this study, ambient aerosol samples collected at the Amazon Tall Tower Observatory (ATTO) during two dry and two wet seasons were characterized by high-resolution mass spectrometry (HR-MS). Comprehensive non-targeted data evaluation was applied to identify thousands of molecular formulae. Most were found to be associated with oxidation products of isoprene and monoterpenes, highlighting the predominance of biogenic secondary organic aerosols (SOA) at ATTO. The chemical composition exhibited distinct seasonal patterns with more processed organic compounds during the dry season, which can be explained by an increase of later-generation oxidation products due to reduced wet deposition and enhanced long-range transport. Mono- and polycyclic heteroaromatic components from biomass burning (BB) sources were enhanced during the dry seasons and the second wet season. The wet seasons were generally characterized by less oxidized compounds, associated with freshly formed SOA particles. Height-resolved measurements showed biogenic emissions with higher concentrations of early terpene oxidation products at lower altitudes. Overall, our results provide new insights into the molecular characteristics and seasonality of organic particulate matter at ATTO, helping to constrain the sources and interactions of aerosols, clouds, and precipitation in the Amazon rainforest.

## 1 Introduction

Organic particulate matter (PM) accounts for up to 90 % of the total atmospheric aerosol mass (Kanakidou et al., 2005; Kroll and Seinfeld, 2008; Jimenez et al., 2009). While primary organic aerosols (POA) are emitted directly into the atmosphere (e.g., combustion of biomass), secondary organic aerosols (SOA) are produced by oxidative processes and transformations of volatile organic compounds (VOC), followed by their oxidation products' nucleation and/or condensation in the atmosphere (Seinfeld and Pankow, 2003). Numerous studies have illustrated the complex chemical composition of organic aerosols covering various compound classes and thousands of different organic species (Tolocka et al., 2004; Zhang et al., 2007; Goldstein and Galbally, 2007; Wozniak et al., 2008; Nizkorodov et al., 2011). Organic aerosols have a substantial impact on climate and human health (Pöschl, 2005; Hallquist et al., 2009). Consequently, clarifying the chemical composition of organic aerosols is an essential task of atmospheric-related studies.

Compounds in OA feature a huge variety of functionalities, polarities, and volatilities at various concentration regimes (Goldstein and Galbally, 2007) which poses major challenges for analytical techniques (Nozière et al., 2015). Traditional gas chromatography and liquid chromatography systems coupled with mass spectrometry (GC/MS, LC/MS) have proven useful in identifying certain OA species (Hoffmann et al., 2011). However, these methods are limited to compounds with specific physicochemical properties and, thus, not adequate to resolve complex organic mixtures (Hildmann and Hoffmann, 2024). Hence, high-resolution mass spectrometry (HR-MS) with enhanced resolving power ($\geq$ 100 000) and high mass accuracy ($\leq$ 5 ppm) has received more and more attention in atmospheric aerosol research (Nozière et al., 2015; Burgay et al., 2023; Xie and Laskin, 2024). The combination with soft ionization methods, such as electrospray ionization (ESI), allows a detailed characterization of complex OA samples (Nizkorodov et al., 2011). Several studies have already proven the potential of HR-MS in smog chamber experiments (Thoma et al., 2022; Frauenheim et al., 2024). Hundreds of different organic species have been detected in SOA samples generated by isoprene and monoterpene oxidation (Tolocka et al., 2004; Bateman et al., 2008; Nguyen et al., 2010). Additionally, OA samples from urban and pristine environments have been analyzed by HR-MS to identify transformation processes and potential sources of aerosol constituents (Wang et al., 2017; Tong et al., 2016; Kourtchev et al., 2013; Hettiyadura et al., 2021; Lin et al., 2018).

The Amazon basin is one of the most pristine ecosystems on Earth, comprising approximately 40% of all tropical forests (Goulding and Barthem, 2003; Baccini et al., 2012). The ecosystem impacts the earth's climate through its role in the hydrological cycle, its role as storage for atmospheric carbon, and the radiative effects of its associated cloud systems. However, the Amazon region is heavily impacted by anthropogenic activities such as deforestation (Andreae et al., 2015). Observations and modelling studies show that climate change and deforestation affect the carbon and hydrological cycles in Amazonia, changing to carbon neutrality and affecting precipitation downwind (Artaxo et al., 2022). Consequently, significant seasonal fluctuations in OA concentrations are observed, influencing the radiation balance and cloud processing (Pöhlker et al., 2023; Pöschl et al., 2010; Artaxo et al., 2013). Despite the Amazon's pivotal role in the Earth's atmosphere, our comprehension of the underlying atmospheric processes remains incomplete. For instance, the role of organic compounds in nucleation, cloud formation processes, or the aging and processing of aerosol components remains unresolved. An understanding of these processes is a fundamental prerequisite for assessing the future of the ecosystem in the context of climate and land use change (Shrivastava et al., 2019). To address such questions, a molecular characterization of organic aerosol in the Amazon rainforest is required.

In this study, LC-HR-MS was applied to analyze aerosol filter samples collected at the ATTO site during multiple campaigns across 2018 and 2019. Organic aerosols were sampled at distinct heights (0 m, 42 m, 80 m, 150 m, and 320 m) to investigate vertical gradients in chemical composition. This height-resolved sampling strategy was designed to capture the influence of biogenic emissions near the forest canopy, photochemical transformation processes above the canopy, and the impact of regionally transported, aged aerosol in the upper boundary layer. In particular, sub-canopy and near-canopy samples reflect

local sources and early secondary organic aerosol formation, whereas the 320 m platform provides insight into regional and long-range transport contributions. These data enable a more detailed separation of local and regional influences on OA composition. It is important to note, however, that vertical differences are strongly influenced by seasonal and diurnal meteorological variability, including boundary layer dynamics, atmospheric mixing, and air mass history. Thus, height-dependent measurements must be interpreted within the broader context of meteorological conditions. Non-targeted high-resolution analytical techniques were employed to characterize the molecular composition of OA in relation to emission sources, (trans)formation processes, and atmospheric conditions. This molecular-level understanding contributes to improved representation of aerosol–cloud–climate interactions in atmospheric chemistry and climate models, particularly for pristine yet rapidly changing tropical regions such as the Amazon.

## 2 Experimental section

### 2.1 Aerosol sampling

Ambient $PM_{2.5}$ aerosol samples were collected in the Amazon rainforest at the Amazon Tall Tower Observatory (ATTO). In total, four measurement campaigns were performed to investigate seasonal variations of the SOA chemical composition. The sampling time covered the wet season 2018 (14/Apr/2018 – 19/Apr//2018), the dry season 2018 (22/Oct//2018 – 31/Oct/2018), the wet season 2019 (04/Mar/2019 – 14/Mar//2019), and the dry season 2019 (20/Sep/2019 – 26/Sep/2019). The research site allows height-resolved measurements on a 325 m tall tower in the central Amazon basin (coordinates: S 02°08.752' W 59°00.335') and is described in detail by Andreae et al. (2015). Shortly, the nearest urban region is Manaus which is located approximately 150 km southwest of ATTO. The station is part of the Uatumã Sustainable Development Reserve (UDSR) and is reached via the roughly 12 km distant Uatumã river. The tower is located at about 120 m above sea level on a plateau that is embedded in a dense, non-flooded forest (terra firme) with adjacent floodplain forests and white-sand soil savannas (campinas) and forests (campinaranas). The area at ATTO is characterized by a canopy height of about 35 m. The meteorological conditions at ATTO depend on the location of the Intertropical Convergence Zone (ITCZ) providing clean and rainy circumstances from February until May and polluted and drier conditions from August to November with strong contributions from anthropogenic deforestation and land-use change (Pöhlker et al., 2019). Relevant meteorological parameters were constantly measured and are shown in the supporting information. The corresponding wind profiles and backward trajectories were generated by the HYSPLIT Trajectory Model (Stein et al., 2015; Rolph et al., 2017).

Ambient PM2.5 filter samples were collected at three different heights at the tower (wet season 2018: 42 m – 15 filters + 2 blank filters, 150 m – 14 filters + 2 blank filters, 320 m – 15 filters + 2 blank filters; dry season 2018 : 80 m - 18 filters + 2 blank filters, 150 m – 18 filters + 2 blank filters, 320 m – 18 filters + 2 blank filters; wet season 2019: 80 m – 22 filters + 2 blank filters, 150 m – 22 filters + 2 blank filters, 320 m – 22 filters + 2 blank filters ; dry season 2019: 0 m – 16 filters + 2 blank filters, 80 m – 14 filters + 2 blank filters, 320 m – 16 filters + 2 blank filters,) using borosilicate glass microfiber filter bonded with PTFE (Pallflex® Emfab, 70 mm diameter). A pre-separator (Digitel, DPM2.3) was set up in front of the filter to only collect the size fraction with aerodynamic particle diameters below 2.5 μm at constant flow rates of 38 L min$^{-1}$. The sampling duration was set to 10 h – 14 h to allow daytime and nighttime related sample collection while ensuring sufficient particle mass on the filter material. Blank filters were collected according to the mentioned procedures but with disabled pumps. In total, 204 filter samples were collected and stored in the freezer at -18 °C before their analysis.

### 2.2 Sample preparation and analysis

One half of each filter sample was extracted three times by adding 1.5 mL of a 9:1 mixture of methanol and water (Fisher Scientific, Optima™ grade). The samples were subsequently agitated for 30 min on a laboratory shaker. The combined liquid phases were filtered afterward through a 0.2 μm PTFE syringe filter (Carl Roth, Rotilabo® KC 94.1) and evaporated

completely by a gentle stream of $N_2$. The residue was then reconstituted in 700 μL of a 9:1 mixture of water and acetonitrile (Fisher Scientific, Optima™ grade).

Ten microliters of each sample were analyzed three times by ultra high-performance liquid chromatography (UHPLC) (ThermoFisher Scientific, UltiMate 3000) coupled to a high-resolution Orbitrap mass spectrometer (MS) (ThermoFisher Scientific, Q Exactive™). A heated electrospray ionization (ESI) source was installed and operated in the negative ionization

mode. The instrument was externally calibrated with a calibration solution (Fisher Scientific, Pierce™) and a 2 mM sodium acetate aqueous solution providing mass accuracies below 1 ppm. The sample analysis was performed in the range of *m/z* 50 – 750 at a mass resolving power of 140 000 at *m/z* 200. The following ESI-MS parameters were used for the measurements: ESI capillary temperature 300 °C; spray voltage −3.2 kV; sheath gas flow 30; auxiliary gas glow 10, S-lens RF level 50%. The UHPLC system was operated with a HSS T3 column (100 × 2.1 mm ID, 1.8 μm, Waters, ACQUITY

UPLC®) and the mobile phase A (water with 2% acetonitrile and 0.04% formic acid) and mobile phase B (acetonitrile with 2% water). The conditions for gradient elution were as follows: 0 – 2 min 0% B, 2 – 9 min linear increase to 30% B, 9 – 15 min linear increase to 95% B, 15 – 17.5 min 95% B, 17.5 – 19 min linear decrease to 0% B, 19 – 21 min 0% B at a constant flow rate of 350 μL min⁻¹.

## 2.3 Non-Targeted Data Evaluation

The LC/MS data were processed by MzMine 2.30 to obtain molecular formulae $C_cH_hO_oN_nS_s$ with the following limitations: $2 \le C \le 40$, $2 \le H \le 100$, $0 \le O \le 40$, $0 \le N \le 4$, $0 \le S \le 2$. Fluorine was not taken into account in the data evaluation, although recent studies show that fluorine compounds in the form of per- and polyfluoroalkyl substances (PFAS) can also be detected in traces in aerosols over tropical rainforests (Kourtchev et al., 2024). The mass tolerance for the formula assignment was set to ± 2 ppm considering isotope patterns. Adducts were removed and only ions present in all three repeated measurements were

retained. The signal intensities per sample were averaged afterward. Blank samples were processed accordingly and subtracted. Only compounds with signal-to-background ratios ≥ 3 were kept for further evaluations.

The number of rings and double bonds was described by the double bond equivalent (DBE) for each assigned formula providing information on the degree of unsaturation and calculated by equation (2.1) (Badertscher et al., 2001):

$$DBE = nc - \frac{n_H}{2} + \frac{n_N}{2} + \frac{n_S}{2} + 1 \qquad (2.1)$$


where $n$ describes the number of the respective element.

Based on the DBE values the aromaticity equivalent (Xc) was calculated by equation (2.2) to identify mono- and polycyclic aromatic compounds (Yassine et al. (2014)):

$$Xc = \frac{3(DBE - (p \cdot n_O + q \cdot n_S)) - 2}{DBE - (p \cdot n_O + q \cdot n_S)} \qquad (2.2)$$


where $p$ and $q$ represent the fraction of oxygen and sulfur atoms that are involved in the π-bond structure of the molecule. Consequently, these values can vary according to different compound classes. While $p = q = 0.5$ is suggested for carboxylic acids, esters, and nitro functional groups, $p$ and $q$ are set to either 0 or 1 for aldehydes, ketones, alcohols, ethers, and nitroso classes. In the present study, $p = q = 0.5$ was used to calculate Xc considering that negative ESI analysis is highly sensitive to

carboxylic acids (Kourtchev et al., 2016; Wang et al., 2017). According to Yassine et al. (2014), Xc ≥ 2.50 and Xc ≥ 2.71 are indicating monocyclic and polycyclic aromatic compounds, respectively.

The carbon oxidation state (OSC) for CHO compounds was calculated following the simplified equation (2.3) (Kroll et al. (2011)):

$$OSC \approx 2 \cdot \frac{n_O}{n_C} - \frac{n_H}{n_C} \qquad (2.3)$$

with the elemental ratios $O/C$ and $H/C$, respectively.

Furthermore, the saturation vapor pressure ($C^0$) has been calculated to predict the volatility of OA compounds according to equation (2.4) Donahue et al. (2011) and Li et al. (2016):

$$log_{10}C^0 = \left(n_C^0 - n_C\right) \cdot b_c - n_O b_O - 2 \cdot \frac{n_C n_O}{n_C + n_O} \cdot b_{CO} - n_N b_N - n_S b_S \qquad (2.4)$$

where $n_C^0$ describes the reference carbon number, $b$ represents the contribution of each element to $log_{10}C^0$, and $b_{CO}$ denotes the carbon-oxygen nonideality. The $C^0$ values have been calculated for compound classes containing C, H, O, N, and S atoms. The parameters for the calculation based on Li et al. (2016) are listed in the supporting information Table S 1.

Kendrick mass (KM) analysis was used to assign homologous series of the detected compounds, which vary only in the number of a defined base unit (Kendrick, 1963; Hughey et al., 2001). In the present study, $CH_2$ was chosen as the base unit by rescaling the exact mass of 14.01565 to 14.00000. $KM_{CH2}$ was calculated according to equation (2.5) for each compound by renormalizing the exact mass scale (Hughey et al., 2001):

$$KM_{CH2} = exact\ mass \cdot \left(\frac{14.00000}{14.01565}\right) \qquad (2.5)$$

Subsequently, the Kendrick mass defect (KMD) was calculated by equation (2.6):

$$KMD_{CH2} = nominal\ KM - KM_{CH2} \qquad (2.6)$$

where $nominal\ KM$ describes the KM rounded to the next integer. Consequently, compounds differing only in their number of $CH_2$ units have the same KMD values. Homologous series can be identified as horizontal lines in KMD plots.

## 3 Results and Discussion

The chemical complexity of the OA samples is highly influenced by stochastic regional events and the respective meteorological conditions between sampling days, such as the height of the mixing layer and the wind direction. In order to take these influences into account, a distinction is made in the present study between background, variable and total organic aerosol compounds. Only compounds that were observed in more than 75 % of all samples based on their presence (identical molecular formula + retention time) were defined as background compounds. They presumably describe the local OA characteristics, as it is assumed that they are not dependent on individual emission events. For the variable OA characteristics, the evaluation was carried out by subtracting the peak areas of the previously determined background compounds from the peak areas of the total number of identified compounds (= background compounds + variable compounds). The variable

compounds are attributed to irregular atmospheric events, presumably caused by different meteorological conditions. Compounds that were only detected once in the respective data set were excluded as they were not considered representative.

### 3.1 Total OA characteristics

All detected compounds were separated by reversed-phase HPLC prior to MS analysis, which increases the number of measurable compounds and reduces the risk of ion suppression in the ESI source. The chemical composition of the SOA samples was mainly influenced by seasonal effects during the measurement campaigns. A total of 2336 molecular formulas could be assigned, of which 699 were in the range of < 250 Da, 1309 between 250 Da and 450 Da, and 328 above > 450 Da. Typical mass spectra are shown in Figure S 20. Molecular weights (MW) in the range below 250 Da could be assigned to the majority of observed substances in both seasons, while the aerosol composition in the dry season additionally showed signals in the oligomeric range between 300 and 450 Da with lower intensity. While several studies have shown that ozonolysis of biogenic VOCs (e.g. α-pinene, β-pinene, isoprene) in smog chamber experiments produces compounds with high molecular weight and increased signal intensities in the oligomeric range (Kourtchev et al., 2014a; Reinhardt et al., 2007), molecules with MW above 450 Da contributed insignificantly to the total number of compounds in the present filter samples.

All filter samples reveal a high chemical diversity with 875 – 1940 unambiguously assigned molecular formulae with a large fraction of isomeric compounds (69% – 85%) (Table I), highlighting the importance of pre-separating the samples by HPLC. The identified molecular formulae were divided into four subgroups according to their elemental composition: CHO, CHON, CHONS, and CHOS. The CHO subgroup is predominant with $(56 \pm 6)$ % in all samples, followed by CHON with $(20 \pm 7)$ %, CHOS $(17 \pm 5)$ %, and CHONS $(7 \pm 1)$ %, although the contribution of sulfur- and/or nitrogen-containing compounds increases at higher MWs. This trend is in good agreement with similar studies from remote environments (e.g. Amazonia, Brazil: CHO[(-)] with 58-63 %, CHON[(-)] with 25-30 %, CHOS[(-)] with 10 %, CHONS[(-)] with 2 %, Kourtchev et al., 2016; Hyytiälä, Finland: CHO[(-)] with $54.8 \pm 2.2$ %, CHON[(-)] with $21 \pm 3$ %, CHOS[(-)] with $16 \pm 3$ %, CHONS[(-)] with $5.4 \pm 2.2$ %, Kourtchev et al., 2013), while studies from a suburban and urban environment revealed enhanced contributions of CHON and CHONS compounds (Pearl River Delta region, China: CHON[(-)] with 34 %, CHONS[(-)] with 8 %, Lin et al., 2012; Cambridge, UK: CHON[(-)] with 33 %, CHONS[(-)] with 21-26 %, Rincón et al., 2012; Shanghai, China: CHON[(-)] with 21-23.7 %, CHONS [(-)] with 11.2-16.6 % , Wang et al., 2017), proving an increased relevance of nitrogen and sulfur chemistry in more polluted areas. The total calculated elemental ratios are highly variable throughout the seasonal measurement campaigns with $0.60 \pm 0.37$ (mean value ± standard deviation of the data set) and $1.47 \pm 0.49$ for O/C and H/C during the dry season and $0.49 \pm 0.32$ and $1.56 \pm 0.45$ during the wet season, respectively. The wide variability indicates a high chemical complexity within the data set. Comparable values were reported from the boreal forest in Hyytiälä, Finland (0.58 and 1.54 for O/C and H/C, respectively) (Kourtchev et al., 2013). Additionally, the elemental ratios obtained in this study are consistent with smog chamber experiments using certain biogenic VOCs for SOA generation, e.g. α-pinene (0.42 – 0.55 for O/C and 1.5 for H/C, Putman et al., 2012) and limonene (0.5 – 0.6 for O/C and 1.5 – 1.6 for H/C, Kundu et al., 2012). Vertical profiles of α-pinene and limonene were measured at ATTO showing different chemical speciations between day and night (Zannoni et al., 2020; Yáñez-Serrano., 2018). The averaged total oxidation states of CHO compounds in the wet seasons ($- 0.681 \leq OSC \leq - 0.486$) are lower than in the dry seasons ($- 0.525 \leq OSC \leq - 0.324$) indicating more processed organic aerosol in the dry seasons than in the wet seasons. Furthermore, the averaged total aromaticity indices ($Xc \leq 2.50$) do not suggest a significant portion of mono- or polycyclic hydrocarbons.

Table I: Summary of all observed MS signals with unambiguous molecular formula assignment for the four measurement campaigns in 2018 and 2019 (wet season = WS, dry season = DS). The signals are divided into four subgroups regarding their elemental composition. The relative contribution of the subgroups was calculated by dividing the number of compounds of the particular subgroup by the total number of compounds. Additionally, the average values[1] of molecular weight (MW), carbon oxidation state ($OS_C$), aromaticity index (Xc), and isomeric fraction are listed.

| Season | Height m | Number of compounds detected | CHO % | CHON % | CHONS % | CHOS % | MW Da | $OS_C$[2] | Xc | Isomers % |
|---|---|---|---|---|---|---|---|---|---|---|
| **WS18** | 42 | 1095 | 54 | 30 | 6 | 10 | 271 | $-0.647$ | 0.872 | 77 |
| | 150 | 875 | 67 | 15 | 7 | 11 | 261 | $-0.606$ | 0.977 | 80 |
| | 320 | 1293 | 43 | 41 | 7 | 9 | 293 | $-0.681$ | 0.822 | 70 |
| **DS18** | 80 | 1856 | 51 | 19 | 7 | 23 | 245 | $-0.324$ | 1.050 | 83 |
| | 150 | 1940 | 51 | 18 | 8 | 23 | 240 | $-0.353$ | 1.051 | 85 |
| | 320 | 1720 | 52 | 18 | 8 | 22 | 245 | $-0.330$ | 1.070 | 84 |
| **WS19** | 80 | 1555 | 55 | 19 | 5 | 21 | 250 | $-0.535$ | 0.907 | 81 |
| | 150 | 1081 | 57 | 18 | 7 | 18 | 252 | $-0.486$ | 0.922 | 81 |
| | 320 | 1287 | 62 | 17 | 6 | 15 | 255 | $-0.558$ | 1.017 | 82 |
| **DS19** | 0 | 1328 | 60 | 17 | 5 | 18 | 237 | $-0.425$ | 0.831 | 69 |
| | 80 | 1225 | 61 | 14 | 6 | 19 | 233 | $-0.439$ | 0.906 | 84 |
| | 320 | 1050 | 62 | 15 | 6 | 17 | 246 | $-0.525$ | 0.975 | 77 |

[1]Average values are calculated based on the molecular composition of each compound.

[2]Only CHO compounds are considered in the calculation of OSC.

## 3.2 Background OA characteristics

A clear seasonality characterizes the molecular background composition of organic aerosols at ATTO. Air masses at ATTO are in response to the annual north-south migration of the ITCZ (Intertropical Convergence Zone) leading to a seasonal alternation between north-easterly winds during the wet seasons and south-easterly winds during the dry seasons (Pöhlker et al. 2019). Seven-day HYSPLIT backward trajectories (Stein et al., 2015) show that air masses need approximately 6 – 7 days from the west coast of Africa to pass the Atlantic Ocean towards ATTO (Figure S4, Figure S5). Higher wind speeds of up to 15 – 20 m s$^{-1}$ were observed during the wet seasons (dry seaons 10-15m s$^{-1}$), leading to a dilution of the particles at ATTO (Figure S3). Comparable to cleaner marine air masses from the ocean, lower background signals are detected during the wet seasons. In contrast, the dry seasons are considerably influenced by biomass burning (Table S3) and other anthropogenic emissions from the eastern regions in Brazil (Artaxo, 2002; Andreae et al., 2015), resulting in higher concentrations of particulate matter (PM) (Figure S6, Figure S7). Concentrations up to 15 µg m$^{-3}$ have been observed during the dry periods while approximately ten times lower concentrations were detected during the wet season.

The chemical composition of background OA also reflects these seasonal changes in meteorological conditions. 72 – 215 organic compounds were detected during the dry seasons, while 28-60 different compounds were observed during the wet seasons (Table S2). The molecular formulae in the wet seasons are predominated by the CHO subgroup (90 ± 7) %, followed by CHOS with (8 ± 7) %, CHONS (1 ± 1) %, and CHON (1 ± 2) %. The dry seasons show a comparable predominance of the

CHO subgroup (93 ± 3) % and an equal contribution of CHONS compounds (1 ± 1) % but an increased fraction of CHON compounds (4 ± 1) % and a decreased fraction of CHOS compounds (3 ± 2) %. Furthermore, the averaged background aromaticity indices (Xc ≤ 2.50) do not suggest a significant portion of mono- or polycyclic hydrocarbons in both seasons.

### 3.2.1 Van-Krevelen diagrams

Van Krevelen diagrams (VK diagrams) are used to visualize similarities and differences, e.g. between measuring locations, measuring heights, daily or seasonal influences. In a VK diagram, the H/C ratio is shown as a function of the O/C ratio for each molecular formula detected in a sample. This helps for example to visualize the degree of oxidation of the particle phase organics as the most oxidized molecules are located at the lower right corner (Zhang et al., 2021). Consequently, the most saturated and reduced species are found in the upper left region. VK diagrams of the wet and dry season 2018 at different sampling heights are shown in Figure 1. The corresponding VK diagrams for the wet and dry season 2019 show very similar patterns and can be found in Figure S9. The signal intensity is proportional to the size of the symbols. However, the signal intensities of the various compounds certainly only provide semi-quantitative results and should therefore only be converted into concentrations to a limited extent, as different substances are detected with different sensitivities due to the selectivity of the electrospray ionization with regard to ion yield. The most intense ion signals are summarized in Table II with their molecular formula, their possible identity and the precursor species.

General: The background compounds in the wet seasons are all non-aromatics with a high O/C-ratio (O/C ≥ 0.5) and either C4-5 or C7-8 backbones, probably resulting from isoprene and monoterpene (mainly α-pinene, β-pinene, and limonene) oxidation (Kleindienst et al., 2007; Nguyen et al., 2010; Worton et al., 2013; Hammes et al., 2019). These precursor compounds are the most prevalent biogenic VOCs in the Amazon rainforest, released by a large diversity of terrestrial vegetation (Guenther et al., 1995; Greenberg et al., 2004; Zannoni et al., 2020). The more complex chemical background of the dry seasons is clearly illustrated in the VK diagram in Figure 1. Especially the dry season 2018 was characterized by a high abundance of signals with H/C ratios ≥ 1.5 and O/C ratios ≤ 0.5. These compounds are commonly attributed to aliphatic species, which can be related to both, biogenic and anthropogenic sources. Simultaneously, several compounds with low elemental ratios (H/C ≤ 1.0 and O/C ≤ 0.5) were identified as background compounds, typically associated with aromatic hydrocarbons (Wozniak et al., 2008; Mazzoleni et al., 2012). Species within this region were exclusively observed during the dry season 2018, where the highest particle concentrations were detected.

Isoprene SOA: .The ion at $m/z$ 149.0455 ($C_5H_{10}O_5$) was observed in all analyzed filter samples with high intensities. This component has been detected in isoprene oxidation chamber experiments and field studies using SOA filter sampling (Krechmer et al., 2015; Chen et al., 2020). $C_5H_{10}O_5$ was identified to be the most dominant species produced during the photooxidation of isoprene hydroxy hydroperoxides (ISOPOOH), which are important products of the isoprene oxidation under low-NO conditions (Krechmer et al., 2015; Paulot et al., 2009; Nagori et al., 2019). As shown in Figure S8, NO and $NO_2$ mixing ratios were generally <0.5 ppb and <1 ppb at 73.3 m and <2 ppb and <0.5 ppb at 0.05 m for the dry season 2018. For the wet seasons 2018 and 2019 NO mixing ratios were <0.25 ppb at 79.3 m and <1.5 ppb at 0.05 m whereas $NO_2$ mixing ratios were <0.05 ppb at 79.3 m and <0.1 ppb at 0.05 m. NOx data for the dry season 2019 are lacking due to instrument issues. These conditions are consistent with what is commonly defined as low-NO regimes in previous chamber studies (e.g., Krechmer et al., 2015; Paulot et al., 2009; Nagori et al., 2019). OH- initiated oxidation of ISOPOOH leads to only 2.5% to organic aerosols including $C_5H_{10}O_5$, while approximately 90% result in the formation of isoprene epoxydiol (IEPOX) and other gas-phase products (Krechmer et al., 2015). However, $C_5H_{10}O_5$ was also observed during isoprene ozonolysis and during isoprene photooxidation under high-NO conditions (Jaoui et al., 2019). Consequently, the latter pathways appear to be the dominant reactions during the dry seasons with increased NO concentrations (Figure S8), while the ISOPOOH-SOA pathway under low-NO conditions is presumed during the wet seasons.

IEPOX-OS: The samples from the 2018 wet season showed the ion at $m/z$ 215.0231 ($C_5H_{12}O_7S$) as the most intense signal related to IEPOX-derived organosulfates (OS) (Surratt et al., 2007; Surratt et al., 2008). It is established that these OSs are formed from the reactive uptake of IEPOX on acidic sulfate particles under low-NO conditions, which is in agreement with our findings. IEPOX-OS was also found on filter samples from the Amazon rainforest in another study (Kourtchev et al., 2016). The authors observed the highest intensities on filters highly affected by surrounding forest fires. Interestingly, the lowest number of active fires was recorded during the wet season 2018 campaign (Table S 3) (INPE - Instituto Nacional de Pesquisas Espaciais, 2020). Sulfate concentrations during the wet seasons are mainly attributed to biogenic and marine sources (Andreae et al., 2015), while increased levels during drier seasons are related to combustion activities (Pöhlker et al., 2018). Sulfur compounds can also be emitted from soils showing higher emissions with higher soil moisture (Pugliese et al., 2023) leading to a significant organosulfur source from the forest itself. This suggests that IEPOX-OS is a relevant SOA constituent, regardless of the current season, and is consistent with the idea of mixed anthropogenic and biogenic sources.

Monoterpene SOA: The wet season 2019 campaign was characterized by intense ion signals at $m/z$ 157.0506 ($C_7H_{10}O_4$) and $m/z$ 171.0662 ($C_8H_{12}O_4$) (Fig S9), attributed to limonene and α-pinene oxidation products, among others (Hammes et al., 2019; Eddingsaas et al., 2012; Thoma et al., 2022; Florou et al., 2024) suggesting that biogenic sources are dominant contributors to SOA loading at ATTO. Furthermore, the intensities of α-pinene oxidation products were higher in the dry than in the wet seasons. This can be explained by increased ambient temperatures (Fig S1; Fig S2) and photosynthetic active radiation facilitating the emission rates of monoterpenes (Guenther et al., 1991; Kesselmeier and Staudt, 1999). This also affects the diel variation with prevalent concentrations during the day.

Aromatics: All detected ions were classified by their aromaticity equivalent Xc according to Yassine et al. (2014), in order to identify mono- and polycyclic aromatic species. While monocyclic structures such as benzene and functionalized derivatives are indicated by values of Xc ≥ 2.50, condensed polycyclic aromatic compounds are described by Xc ≥ 2.71. Naphthalene is the smallest polycyclic structure meeting this criterion (Yassine et al., 2014; Wang et al., 2017). The dry season campaigns were accompanied by an increased number of active fires and biomass burning (Table S 3). Consequently, the detected aromatic hydrocarbons are mainly attributed to anthropogenic sources (Henze et al., 2008).

Hydrocarbons: The dry season 2018 was characterized by the occurrence of several aromatic species with Xc values ≥ 2.50 (Figure S8). The most intense signals correspond to the ions at $m/z$ 149.0243 ($C_8H_6O_3$), $m/z$ 164.0353 ($C_8H_7NO_3$), $m/z$ 193.0505 ($C_{10}H_{10}O_4$), and $m/z$ 207.0298 ($C_{10}H_8O_5$), each with DBE values ≥ 6. These compounds were already identified as relevant components of biomass burning activities. $C_8H_6O_3$ and $C_{10}H_8O_5$ were observed during naphthalene photooxidation experiments in a chamber study (Kautzman et al., 2010; Chhabra et al., 2015; Thoma et al., 2022). Naphthalene and other polycyclic aromatic hydrocarbons (PAH) have been related to wood combustion processes in the literature and are considered to impact human health (Schauer et al., 2001; Chan et al., 2009; Samburova et al., 2016; Baumann et al., 2023). $C_{10}H_{10}O_4$ might be a product formed during the degradation of cellulose and lignin, two important biopolymers (Kong et al., 2021). The MS$^2$ spectrum (Figure S11) revealed two fragments, $CO_2$ and $C_3H_4O_2$ (acrylic acid), leading to a tentative structure of ferulic acid.

Nitro-hydrocarbons: $C_8H_7NO_3$ was assigned to a nitro-phenolic structure, which can be formed during biomass pyrolysis. This compound class is a major contributor to brown carbon (BrC) (Lin et al., 2016) and has been detected earlier in SOA samples from the Amazon rainforest (Claeys et al., 2012). The ion appeared in almost every sample of the dry season 2019 in high intensities, suggesting this compound to be an important biomass burning marker in this study. Another intense background ion was 4-nitrocatechol at $m/z$ 154.0146 ($C_6H_5NO_4$), which has been previously considered as a marker compound for biomass burning OA (Iinuma et al., 2010; Claeys et al., 2012; Kourtchev et al., 2016).

Polycycles: The ion at $m/z$ 219.0455 ($C_{15}H_8O_2$) appeared in the dry season 2019 samples and has already been reported in another study (Bruns et al., 2015). The authors identified this compound as an oxygenated PAH produced during wood combustion with a pyrene core structure, which seems reasonable according to the increased Xc Value of 2.82 and 12 DBE.

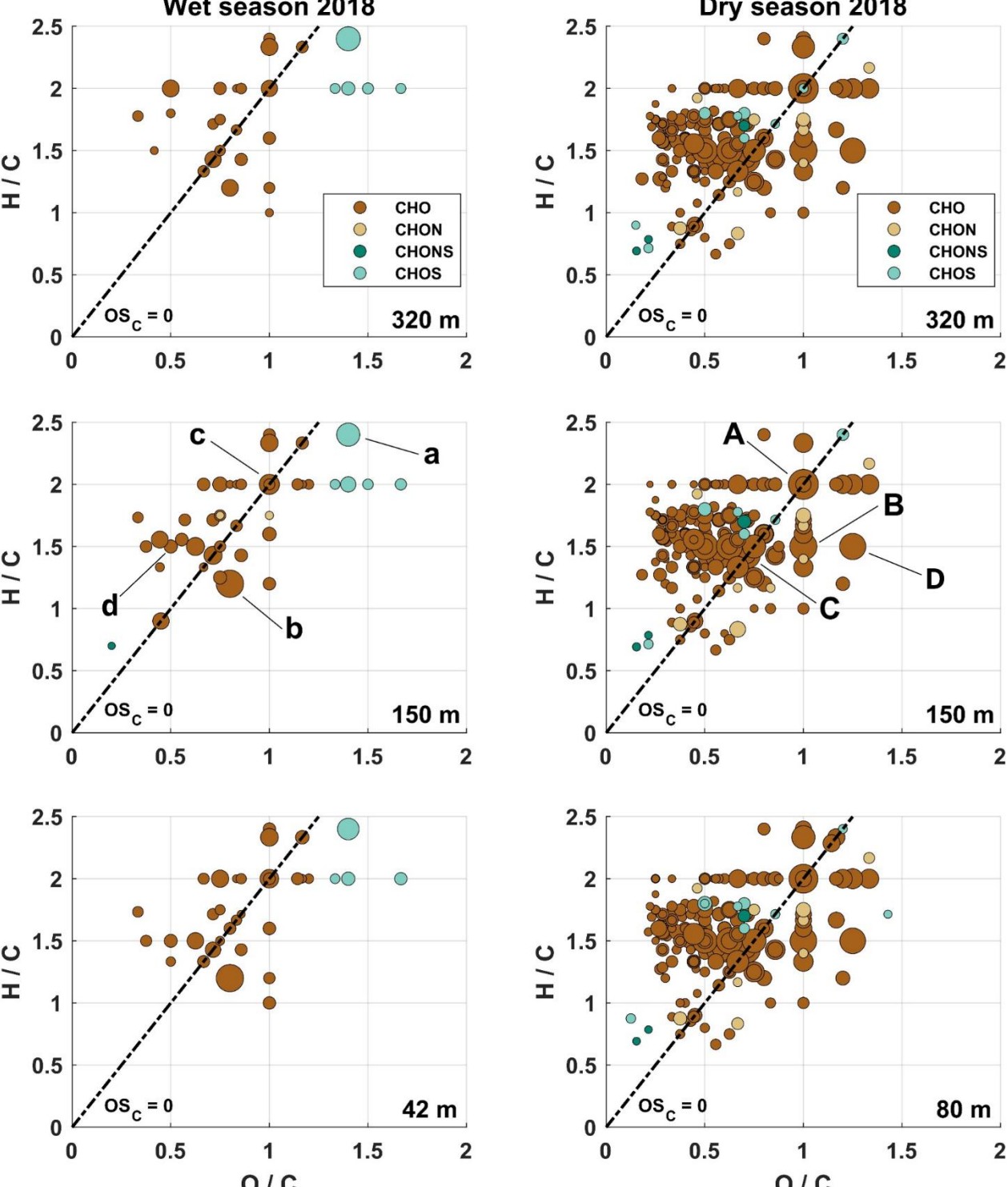

**Figure 1:** Van Krevelen plots from the wet season 2018 (left panel; sampling heights: 320 m, 150 m, 42 m) and the dry season 2018 (right panel; sampling heights: 320 m, 150 m, 80 m). Included are only molecular formulae that were present in more than 75% of the samples, respectively. The size of the data points represents the signal intensity of the corresponding peak. The four subgroups are distinguished with different colors. Compounds located on the black dashed line have an average carbon oxidation state of 0 ($OS_C = 0$). The labels a-d and A-D correspond to the most intense signals further characterized in Table II.

**Table II: Possible identities and precursors for the most intense signals from the background ions for the different seasons.**

| Season | ID[1] | Formula | *m/z* | OSC | Possible identity | Precursor | Reference |
|---|---|---|---|---|---|---|---|
| **WS18** | a | $C_5H_{12}O_7S$ | 215.0231 | - | IEPOX-OS | Isoprene | (Worton et al., 2013) (Kourtchev et al., 2016) |
| | b | $C_5H_6O_4$ | 129.0194 | 0.4 | Dicarboxylic acid / Peroxy acid | Methylfuran | (Joo et al., 2019) |
| | c | $C_5H_{10}O_5$ | 149.0455 | 0.0 | Carbonyl-tetrol / Epoxyl-tetrol / Carboxyl-triol | Isoprene | (Krechmer et al., 2015) (Chen et al., 2020) |
| | d | $C_8H_{12}O_4$ | 171.0662 | −0.5 | Ketolimonalic acid / Terpenylic acid | Limonene / α-pinene | (Hammes et al., 2019) (Eddingsaas et al., 2012) |
| **DS18** | A | $C_5H_{10}O_5$ | 149.0455 | 0.0 | | See ID c | |
| | B | $C_4H_6O_4$ | 117.0193 | 0.5 | Succinic acid | - | (Wang et al., 2017) |
| | C | $C_7H_{10}O_5$ | 173.0454 | 0.0 | | α-pinene | (Chen et al., 2020) (Kleindienst et al., 2007) |
| | D | $C_4H_6O_5$ | 133.0142 | 1.0 | Malic acid | Isoprene | (Nguyen et al., 2010) |

[1] IDs correspond to the labels illustrated in Figure 1.

### 3.2.2 Carbon oxidation states

355 To further characterize the background chemical composition of the aerosol, the average carbon oxidation state (OSC) was calculated for each detected molecular CHO formula. This metric was introduced by Kroll et al. (2011) and is a useful parameter to describe complex mixtures of organic aerosols. Various chemical and physicochemical processes in the atmosphere are determined by the oxidation state of the organic components. For example, highly oxidized molecules (HOM) with sufficiently low volatility can form clusters, which leads to the formation of new particles and subsequent particle growth,
360 even of small particles (Bianchi et al., 2016; Molteni et al., 2016).

To give an impression of the position of the corresponding OSC values within the regular Van Krevelen plots, organic compounds with OSC = 0 are indicated by the dashed line in Figure 1. To the left are compounds with an OSC <0 (not or less oxidized), to the right are compounds with OSC values > 0, e.g. the HOMs, which are defined as molecules with O/C ≥ 0.6 and/or OSC ≥ 0 (Tu et al., 2016). As shown in Figure 1, the samples from the 2018 dry season had the highest number of
365 highly oxidized compounds, which accounted for about 30 % of the total number of compounds. The average molecular formula for the HOMs in the 2018 dry season was C6.5H8.6O5.1 with C4-C10 backbones, which can be interpreted that isoprene and monoterpene precursors are also the main source of HOMs in the Amazon rainforest in the dry season. The 2019 dry season showed fewer oxidized molecules, with a mean molecular formula of C6.7H10.2O5.7. The difference in HOM concentrations between the dry seasons of 2018 and 2019 may be related to differences in the chemical composition of OH
370 reactivity during those periods. Pfannerstill et al. (2020) observed that while total OH reactivity was higher in September 2019 than in October 2018 (29.1 ± 10.8 s⁻¹ vs. 28.1 ± 7.9 s⁻¹ at 80 m), this increase was not driven by monoterpenes, which are key precursors for HOM formation. The high OH reactiviy in the dry season 2019 was rather dominated by oxidized VOCs, such as aldehydes and organic acids. This shift toward more oxidized and aged VOCs indicates a more photochemically processed

air mass, potentially affected by biomass burning. Although NOx was not measured (Figure S8), the authors suggest that fire-related emissions may have altered the oxidation regime. Given that high NOx levels can suppress $RO_2$ autoxidation and thereby inhibit HOM formation, and that monoterpene contributions to OH reactivity were not elevated in 2019, it is plausible that the observed reduction in HOMs was due to a combination of reduced precursor availability and a shift in chemical conditions unfavorable for HOM production.

However, it should be noted that the analysis of highly oxygenated compounds would benefit from the use of a more polar stationary phase of the LC column, as a more comprehensive analysis of analytes of the entire polarity range would be enabled. This would ensure that even the very polar species are not underestimated due to insufficient separation.

Figure 2 shows the OSC values as a function of carbon number for each CHO compound, with OSC values ranging from -1.5 to +1.2 in the dry seasons and from -1.1 to +1.0 in the wet seasons. This result is consistent with other studies in remote continental areas (Kourtchev et al., 2013; Kourtchev et al., 2016). Molecules with an OSC between -0.5 and -1.5 and a carbon number greater than seven are considered biomass-burning organic aerosol (BBOA). The comparison between dry and wet seasons clearly shows the large number of compounds associated with BBOA and that these substances contribute substantially to the SOA load of the Amazon rainforest during dry seasons. Interestingly, the contribution of BBOA-related compounds was significantly higher in the 2019 wet season (Figure S 12), which is likely due to the increased number of active fires during this period. In contrast, molecules with an OSC between -0.5 and +1.0 and less than 13 carbon atoms are generally associated with semivolatile and low-volatility oxygenated organic aerosol (SV-OOA and LV-OOA) (Kroll et al., 2011; Wang et al., 2017). These compounds are formed by multiple oxidation reactions and are commonly attributed to fresh and aged SOA, respectively (Zhang et al., 2007; Jimenez et al., 2009). The background chemical composition during the dry seasons also revealed numerous compounds attributed to LV-OOA indicating more aged SOA components compared to the wet seasons. Whether the lower precipitation in the dry season and the associated longer residence times of the OA in the atmosphere are the cause of these observations, or whether these compounds also have their source in biomass combustion processes, cannot be clearly determined here. In contrast, the majority of background compounds during rainy seasons are associated with SV-OOA, suggesting that local biogenic emissions are the main source of freshly formed SOA. Overall, Figure 2 clearly shows that large differences in chemical composition exist between the two seasons in the Amazon rainforest.

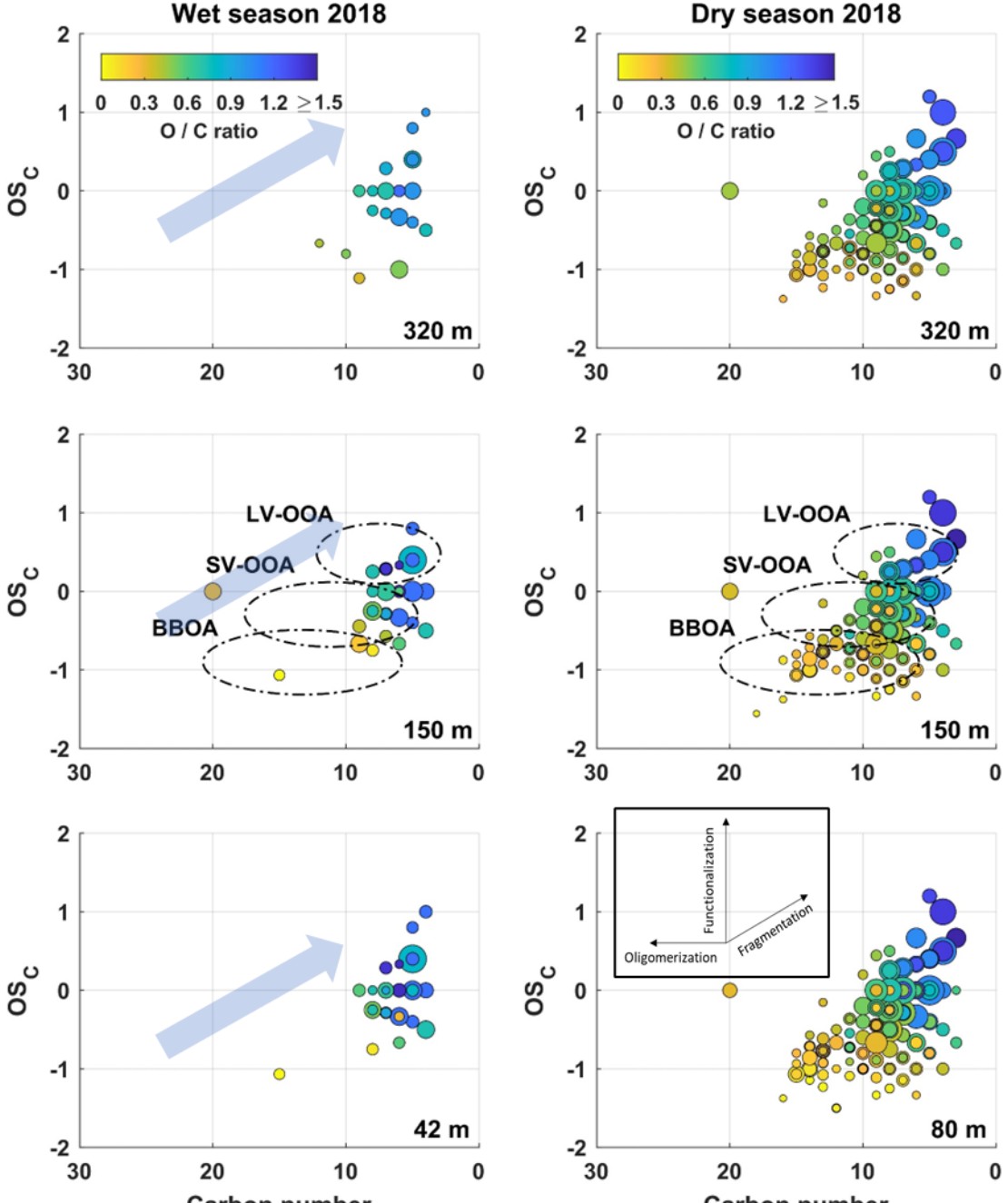

**Figure 2: Carbon oxidation state (OSC) plots for all detected CHO species during the wet season 2018 (left) and dry season 2018 (right) at different sampling heights. Only background ions are included. The size of the data points represents the signal intensity of the corresponding peak. The color code illustrates the degree of oxygenation. The black dashed areas are related to low-volatility (LV-OOA) and semivolatile (SV-OOA) oxygenated organic aerosol and biomass-burning organic aerosol (BBOA). Insets: vectors corresponding to key classes of reactions of atmospheric organics: functionalization (addition of polar functional groups), fragmentation (cleavage of C–C bonds), and oligomerization (covalent association of two organic species). The combination of these reaction types leads to complex movement through (-the OSC–carbon number-) space; however, the inevitable increase in OSC with atmospheric oxidation implies that, given enough time, organics will generally move up and to the right (blue arrows). Insets based on Kroll et al., 2011.**

### 3.2.3 Height-dependency of the background OA characteristics

The vertical profiles of the background organic aerosol (OA) composition exhibited only subtle variations across the sampled heights, with much smaller gradients than those observed between seasons. As illustrated in the van Krevelen diagrams (Figure 1) and summarized in Table S2, the background OA was dominated by CHO-type compounds at all heights and during all seasons, accounting for 82 % to 98 % of the detected molecular formulas. Nitrogen- and sulfur-containing species (CHON,

CHOS, CHONS) contributed only minor fractions, with a slight increase in CHON and CHONS observed at 150 m during the wet season 2018 (CHON: 5 %, CHONS: 2 %). During the wet season 2019 CHON and CHONS species accounted for 0 % and 2 % at all sampling heights (80 m, 150 m, 320 m). The OSC of the background CHO species showed a weak increase with altitude during the wet season 2018 (Fig 2). At 320 m, the mean OSC of CHO compounds was slightly higher (+0.04) compared to 150 m (−0.04) and 42 m (+0.03, Table S2), suggesting that the aerosol at higher altitudes had undergone marginally more extensive oxidation, likely due to longer atmospheric residence times or vertical transport from above the forest. Contrastingly, during the wet season 2019 the mean OSC of CHO compounds did not show significant variations at different sampling heights (around -0.5, Table S2). The magnitude of order for the OSC values for the wet season 2019 were comparable to the dry season 2018. Hereby should be noted that the sampling heights from the wet season 2019 and dry season 2018 (80 m, 150 m, 320 m) differed from the wet season 2018 (80 m instead of 42 m) and the dry season 2019 (0 m, 80 m, 320 m). Therefore, the chemical composition in the wet season 2018 could also include more local OA sources at the canopy level (sampling at 42 m) whereas the dry season 2018 and wet season 2019 probably reflect more regional sources that are well mixed. However, during the dry season 2018 and wet season 2019, vertical gradients nearly disappeared. The background CHO compounds at all heights exhibited similar OSC values (around –0.4 for the dry season 2018 and -0.5 for the wet season 2019, Table S2) with a significant lower number of signals in the wet season 2019 (51-60) than in the dry season 2018 (209-211). The compounds clustered closely in the SV-OOA and LV-OOA regions (Fig 2), suggesting the dominance of regionally aged aerosol, as well as pronounced BBOA signals originating from biomass burning and other long-range sources. The dry season 2019 exhibited a markedly lower contribution of biomass burning organic aerosol (BBOA) signals (Fig S12) and a reduced number of detected molecular formulas (72-198, Table S2) compared to the dry season 2018 (209-211, Table S2). Despite similar average oxidation states (OSC ~ –0.4), the aerosol composition in the dry season 2019 was characterized by a lower contribution of aromatic compounds ($X_c$ = 0.3–0.5) and sulfur-containing species (CHOS 1 % at 150 m). These differences suggest a shift in the dominant sources and atmospheric processes during this period. Interestingly, fire activity across the Amazon basin was in fact higher in 2019, with 19925 recorded fires (Table S3), compared to previous years. However, the expected increase in biomass burning tracers at ATTO was not observed. This discrepancy suggests that the smoke plumes generated by these fires were either transported in other directions or did not reach the ATTO site efficiently. Possible explanations include altered large-scale atmospheric circulation patterns, such as changes in wind direction or vertical stability, which limited the transport of biomass burning emissions from southern and southwestern Amazonia towards the northeastern Amazon, where ATTO is located. In the reduced biomass burning influence, the aerosol composition at ATTO during the dry season 2019 was shaped predominantly by regional background air masses and local processes. The highest oxidation state of CHO-type compounds was observed below the canopy at 0 m sampling height (OSC: –0.367), along with the highest mean molecular weight (196 Da). This points to local oxidative processes near the forest floor, such as the oxidation of biogenic emissions from soil and understory vegetation or nighttime nitrate radical chemistry. Limited vertical mixing during stable nighttime conditions may have allowed the accumulation of heavier, more oxidized compounds below the canopy.

In total, the observed vertical differences in the background composition are subtle and become significant only under low-turbulence wet season conditions like in the wet season 2018. Even then, the background aerosol remains compositionally similar across heights, reflecting the strong influence of regional air masses and long-range transport on the background aerosol burden at the ATTO site. These findings highlight that, while vertical gradients in the background aerosol composition are generally small when averaged over entire seasons, individual events or specific atmospheric conditions may exhibit much more pronounced vertical or temporal variability. Therefore, future studies will focus on single event-based analyses examining specific episodes of chemical stratification, nighttime oxidation, or pollution influence that could provide a more detailed understanding of the underlying processes.

### 3.3 Variable OA characteristics

The occurrence of variable component contributions to the chemical composition of OA is attributed to irregular atmospheric events, which are presumably caused by varying meteorological conditions, i.e. uncommon wind direction, long-range transported plumes or fires or anthropogenic pollution nearby. Compounds that were only detected once in the respective data set were excluded, as they were not considered representative. As already mentioned, the previously discussed background compounds were not taken into account for the group of variable compounds defined here. The remaining signals were assigned according to the sampling time during the day (7:00 - 17:00 local time, UTC-4 h) and night (17:00 - 7:00 local time, UTC-4 h). An additional sample in the morning (7:00 – 12:00 local time, UTC-4 h) was collected during the dry season 2019 only, while the daytime sample was collected between 12:00 and 17:00 local time, UTC-4 h. The results and trends for 2018 and 2019 are discussed in this chapter, but for clarity, the graphs for 2019 are shown in the supplement (Figure S 13-16).

### 3.3.1 Molecular tracer compounds of individual events

In this section, Van Krevelen and OSC diagrams are discussed with the aim of identifying molecular tracer components that can be associated with individual meteorological or atmospheric chemical events. Representative VK diagrams for the seasons in 2018 are shown in Figure 3. The respective diagrams of the wet and dry season 2019 are shown in Figure S13 and S14. For better understanding, the VK diagrams were divided into five areas: *I* for very highly oxidized organic compounds, *II* for highly oxidized compounds, *III* for intermediately oxidized compounds, *IV* for oxidized unsaturated organic compounds and *V* for highly unsaturated organic compounds. The five areas were defined according to Zhang et al., 2021.

(*I*) Very highly oxidized organic compounds: Area *I* shows very highly oxidized organic compounds, with isoprene oxidation products being a distinctive feature. Since less wet deposition of aerosol particles occurred due to less precipitation events during the dry season 2018, a greater number of compounds were identified during the dry season 2018 than during the wet season 2018. This result is consistent with the particle mass and number concentrations (Figure S 6, Figure S 7) observed during these campaigns. $C_5H_{12}O_4$ (*m/z* 136.0662) was detected in all samples from all seasons and identified as 2-methyltetrole, a prominent product of isoprene oxidation (Claeys et al., 2004). A large number of CHOS compounds were detected during the dry season 2018. More than 85 % of the CHOS molecules detected showed an O/S ratio $\geq$ 4, indicating the presence of at least one sulfate functional group. Species with O/C $\geq$ 1 and H/C $\geq$ 2 with highly oxygenated C4 and C5 backbones were particularly prominent. Their structures emphasize that isoprene is the main precursor for S-containing OA at ATTO. $C_5H_{12}O_7S$ (*m/z* 215.0231, IEPOX-OS) and $C_5H_{10}O_7S$ (*m/z* 213.0075) were detected in filter samples from all four campaigns and showed the highest intensities. Both structures have already been identified as isoprene-derived OS and correlate with increased sulfate concentrations during dry periods. The latter can be attributed to the long-range transport of anthropogenic emissions (Andreae et al., 2015). It is well established that isoprene-derived organosulfates are primarily formed via photochemical oxidation mechanisms, involving reactive intermediates such as ISOPOOH and IEPOX under low-NO conditions, often leading to enhanced production during the daytime (Surratt et al., 2007; Surratt et al., 2008). However, in our study, we observed consistently higher CHOS signal intensities at night across all seasons. While OS formation is photochemically driven, the nocturnal enhancement in signal intensities is likely not indicative of in situ nighttime production, but instead reflects more efficient partitioning into the particle phase during cooler nighttime conditions. This explanation is consistent with the findings of Gómez-González et al. (2012) and Kourtchev et al. (2014b), who also reported strong diurnal differences driven by temperature-dependent gas-particle partitioning.

Concerning the height-dependent analysis the wet season 2018 reveals that CHON and CHOS species in area *I* were more abundant at 42 m (slightly above canopy) during the day, whereas their number increased at 320 m during the night. CHO compounds, in contrast, showed a lower relative contribution at 320 m compared to 42 m in both, day and night samples. (Fig 3). These observations suggest a vertical redistribution of specific compound classes influenced by boundary layer dynamics. The nocturnal increase in CHON and CHOS compounds aloft may be related to residual layer effects or long-range

transport, whereas the canopy-level daytime maxima point to local, biogenic emissions and in-canopy photochemical processing. In the dry season 2018, no clear vertical trend was observed in area *I*, suggesting enhanced atmospheric mixing and a more homogeneous chemical distribution due to convective processes. These interpretations are supported by recent findings that showed that under very stable nocturnal conditions (assumed for wet season 2018), the boundary layer height can remain below 100 m, allowing upper levels (e.g., 320 m) to remain decoupled and enriched in aged or transported aerosol components. Furthermore, the prevalent southwesterly wind direction in the dry season 2018 is associated with higher surface roughness and increased turbulence, potentially leading to more uniform vertical distributions and mixing of aged, oxidized species (Mendonca et al., 2025).

(*II*) Highly oxidized organic compounds: In general, area *II* is characterized by highly oxidized organic species like late monoterpene oxidation products due to oxidative aging or higher generation oxidation products (Zhang et al., 2021). The dry season 2018 shows a higher number of compounds than the wet season 2018, indicating more aged organic aerosol. This is also underlined by an increased number of LV-OOA species in the OSC plots in Figure 4. An intense ion signal with *m/z* 203.0562 ($C_8H_{12}O_6$) could be assigned to 3-methyl-1,2,3-butanetricarboxylic acid (MBTCA) by comparison with an authentic standard (Table S5). MBTCA is a well-established tracer compound for aged biogenic SOA formed by the photooxidation of α- and β-pinene (Szmigielski et al., 2007; Zhang et al., 2010). Similarly, two detected CHONS compounds at *m/z* 294.0654 ($C_{10}H_{17}O_7NS$) and *m/z* 296.0446 ($C_9H_{15}O_8NS$) were associated with aged SOA, which were previously identified as oxidation products of α-pinene, β-pinene and limonene (Surratt et al., 2008). The O/(N+S) ratio $\geq 3.5$ allows for the presence of both -$OSO_3H$ and -$ONO_2$ functional groups and assignment as nitrooxy organosulfates (NOS). Similar to the higher concentrations of very highly oxidized CHOS species (*I*), highly oxidized (*II*) CHONS species also show increased concentrations in dry periods and high correlations with $SO_2$ levels, but also $NO_x$ concentrations (increased $NO_x$ and black carbon equivalent concentrations (BCe) in the dry season 2018; Figure S8 and S17). This underlines the importance of anthropogenic emissions reaching the ATTO site by long-distance transport and changing the chemical composition of organic aerosols by mixing biogenic and anthropogenic sources.

The molecular assignment of two further ions, *m/z* 161.0455 ($C_6H_{10}O_5$) and *m/z* 129.0194 ($C_5H_6O_4$) with increased concentrations in the dry season clearly shows a connection with biomass burning activities. The overall burning activity in the basin was significantly lower in the wet season 2018 compared to the dry season 2018 (Table S 3). $C_6H_{10}O_5$ can be assigned to 1,6-anhydro-β-D-glucopyranose (Levoglucosan) by comparison with authentic standard substances (Table S5). Levoglucosan is a generally recognized product of cellulose pyrolysis that is used as a marker compound for biomass burning activities (Simoneit et al., 1999; Nolte et al., 2001). However, $C_5H_6O_4$ can be explained less clearly. A compound with this molecular formula was found as an oxidation product of the reaction of 3-methylfuran with $NO_3$ radicals (Joo et al., 2019), while furans themselves were detected as products of cellulose combustion (Mettler et al., 2012). The $C_5H_6O_4$ molecule observed here showed higher concentrations in dry season samples (higher number of fires) and in night samples (nitrate radical chemistry), supporting the hypothesis that the reaction of 3-methylfuran with $NO_3$ radicals is a source of this compound. However, the C5 backbone can also suggest isoprene as potential precursor, which can also form $C_5H_6O_4$ according to photooxidation experiments in chamber studies (Nguyen et al., 2011; Clark et al., 2013).

During the wet season 2018, the oxidized organic aerosol subtype LV-OOA showed its highest O/C ratios at 150 m during daytime (Fig 4), indicating a vertical gradient in oxidation state. While the oxidation state increased with height, the total number of LV-OOA species decreased at 320 m. This pattern is consistent with oxidative aging during vertical transport and lower precursor availability aloft. At night, this vertical pattern persisted, albeit with overall lower O/C ratios (Fig 4). CHON signals in area *II* were most abundant at 320 m, leading to the assumption that highly oxidized, aged CHON compounds reached the ATTO site by long-range transported aerosols.

(*III*) Intermediately oxidized organic compounds: Area *III* mostly consists of first-generation oxidation products of monoterpenes and sesquiterpenes that are intermediately oxidized (Zhang et al., 2021). Again, the dry season 2018 is characterized by a higher number and concentration of compounds than the wet season 2018 while no significant differences are observed between the daytime and nighttime samples. The compounds are characterized by aliphatic species that can be attributed to mixed anthropogenic and biogenic sources (Henze et al., 2008; Kourtchev et al., 2013). A prominent ion was

observed at *m/z* 185.0819, which was assigned to the molecular formula $C_9H_{14}O_4$. The compounds with this molecular formula showed multiple signals in the LC-MS analysis, indicating the presence of several isomeric structures. One signal was identified as pinic acid (PA) from an authentic standard compound (Table S5). The additional signals are likely to be oxidation products of other monoterpenes, such as β-pinene, limonene and $\Delta^3$-carene (Jenkin, 2004; Chen and Griffin, 2005; Hammes et al., 2019). In contrast to MBTCA, pinic acid is an oxidation product that is already formed during the first oxidation steps

of α-pinene and can therefore be used as a tracer for freshly formed SOA. Later generation products, such as MBTCA, were more concentrated in the dry seasons, indicating more processed SOA particles, while PA was predominant in the wet periods. The OSC analysis in Fig 4 shows that the dry season 2018 has pronounced signals at small numbers of C atoms with high O/C ratios, whereas the wet season 2018 is characterized by pronounced SV-OOA species, supporting the different seasonal variation of different generations.

During the wet season 2018, CHO and CHON species in area *III* were most abundant at 42 m during daytime, indicating a dominant contribution from local biogenic sources within or just above the canopy. CHO compounds decreased with height, while CHON compounds remained relatively constant (Fig 3). At night, however, CHON signals dropped at 42 m, possibly due to suppressed vertical mixing or altered chemical production pathways. This suggests that canopy dynamics and photochemical activity strongly influence the vertical distribution of first-generation monoterpene oxidation products. These

trends are consistent with the findings of Mendonça et al. (2025), which indicate that under wet season conditions, when the wind predominantly arrives from the northeast (Fig S5) and the surface roughness is relatively low, the nocturnal boundary layer is shallow (typically 80–120 m), limiting vertical exchange and favoring accumulation of semivolatile species near the canopy top.

       (*IV*) Oxidized unsaturated organic compounds: This area is mainly characterized by unsaturated hydrocarbons with a low

oxygen content. Most of these compounds are classified as CHON, CHOS and CHONS with Xc ≥ 2.5, indicating aromatic molecular structures. The dry season 2018 shows a higher diversity of compounds whereas the wet season 2018 consists of a lower number of compounds mainly dominated by CHO species. However, the dry season of 2018 shows a higher presence of CHON species during nighttime compared to daytime samples. The most intense ion signals in the dry season 2018 *m/z* 154.0146 ($C_6H_5NO_4$, 4-nitrocatechol), *m/z* 168.0301 ($C_7H_7NO_4$, methyl-nitrocatechols) and *m/z* 185.0457 ($C_8H_9NO_4$,

dimethyl-nitrocatechols) can be associated with nitroaromatic compounds, which are also considered markers for biomass burning OA (Iinuma et al, 2010; Claeys et al, 2012; Kitanovski et al, 2012; Kahnt et al, 2013; Siemens et al., 2023). Kourtchev and coworkers (Kourtchev et al., 2016) also observed elevated concentrations of these compounds in the Amazon region during the period when numerous forest fires occurred. In contrast, no nitrocatechols were detected for the wet season 2018. However, $C_6H_5NO_4$ was the only compound identified as a background species during the dry season 2018 (Figure 1), suggesting a lower

SOA contribution from $C_7H_7NO_4$ and $C_8H_9NO_4$ at ATTO.

       Vertically resolved analysis of the wet season 2018 revealed that CHO signals in area *IV* were less abundant at 320 m compared to lower levels, similar as described in area *III*. No significant day-night differences were observed, indicating that these compounds may be governed more by long-range transport and vertical diffusion than by in situ photochemistry. The lack of clear diurnal variation points to a more persistent source, possibly influenced by combustion-related emissions or stabilized

air masses.

       (*V*) Highly unsaturated organic compounds: Area *V* is predominantly composed of highly unsaturated organic compounds that are combustion-related (Zhang et al., 2021). The number of compounds and the variety of species is enhanced for the dry

season 2018 compared to the wet season 2018. This trend is also reflected in the BBOA areas in Figure 4. Furthermore, black carbon equivalent concentrations (BCe) were enhanced during the dry season 2018 compared to the wet season 2018 (Figure

S17). No single dominant species was identified among the combustion-related highly unsaturated organic compounds, as their composition varied substantially between seasons. While the wet season 2018 shows mainly CHO species with a decreased number of compounds during nighttime, the dry season 2018 is dominated by CHOS and CHONS species and an increased number of CHON compounds during nighttime, suggesting enhanced particle-phase partitioning at lower temperatures.

During the wet season 2018, a greater number of highly unsaturated compounds were detected at 42 m compared to 320 m, both day and night. This points to a near-surface deposition and limited vertical mixing under stable boundary layer conditions, whereas no significant height resolved differences were observed for the dry season 2018.

The wet season of 2019 stands out due to individual pollution events that significantly influenced aerosol composition at the
ATTO site. The number of components and their intensity increase across areas *I–V* for the wet season 2019 (Figure S13), compared to the wet season 2018 (Figure 3) and the dry season of the same year (Figure S14). The wet season 2019 also shows higher levels of CHON and CHO species relative to the 2019 dry season. The concentrations of OS related to isoprene remain comparable to those observed during the dry seasons of 2019 and 2018. In particular, area *V* in Fig S13 and the BBOA area in Fig S15 highlight the impact of biomass burning during this period. This is further supported by elevated BCe concentrations
(Fig S17). While the background OA characterization still shows typical wet season features, such as low BBOA contributions, a dominance of early-stage monoterpene oxidation products, minimal aging effects (Figure S12), and absent signals in area *V* (Figure S9), it becomes evident that, with the inclusion of individual events, episodes of considerable pollution were present, temporarily altering the aerosol profile. These deviations likely reflect enhanced fire or combustion activity upwind or near the ATTO site during this period (Table S3). Additionally, emissions from biomass burning in the Sahel savanna regions,
transported across the Atlantic Ocean (Fig S5), could have contributed to the observed aerosol characteristics (Holanda et al., 2020; Holanda et al., 2023).

In the wet season 2019, no significant vertical trends were observed across areas *I–V*, indicating a well-mixed atmosphere, possibly due to convective activity or large-scale transport. However, the dry season 2019, which included sampling at three different times and at sub-canopy level (0 m), revealed unique patterns. SV-OOA and CHOS compounds in areas *I* and *II*
peaked at 80 m during nighttime, consistent with gas-particle partitioning favored by cooler, stable layers above the canopy. CHON compounds were most abundant at 0 m and 80 m in the morning but diminished at 320 m. These patterns underline the importance of both vertical stratification and local chemical processing in modulating aerosol composition at different heights and times of day. Mendonça et al. (2025) noted that dry season nights at ATTO, characterized by southwesterly winds and enhanced surface roughness, can exhibit deeper but more turbulent boundary layers, allowing complex layering and
submesoscale motions to form, which could explain the varied height-dependent signals observed in this campaign. Moreover, the presence of the highest CHOS and SV-OOA signals at 80 m during nighttime suggests a zone of active condensation and SOA formation just above the canopy, where gas-particle partitioning is favored by cooler and more stable stratification. The fact that CHON species were relatively suppressed at 320 m both during morning and daytime indicates limited upward transport of nitrogen-containing precursors or their rapid transformation near the surface. The chemical differences between
0 m and 80 m, particularly for CHON and CHO species in areas *II* and *III*, also suggest that in-canopy processes such as deposition, emissions, and light penetration significantly modulate chemical composition. Together, these observations highlight the sensitivity of nighttime aerosol chemistry to fine-scale vertical structure and suggest that sub-canopy and canopy-top levels may act as chemically distinct compartments in the nocturnal boundary layer during the dry season.


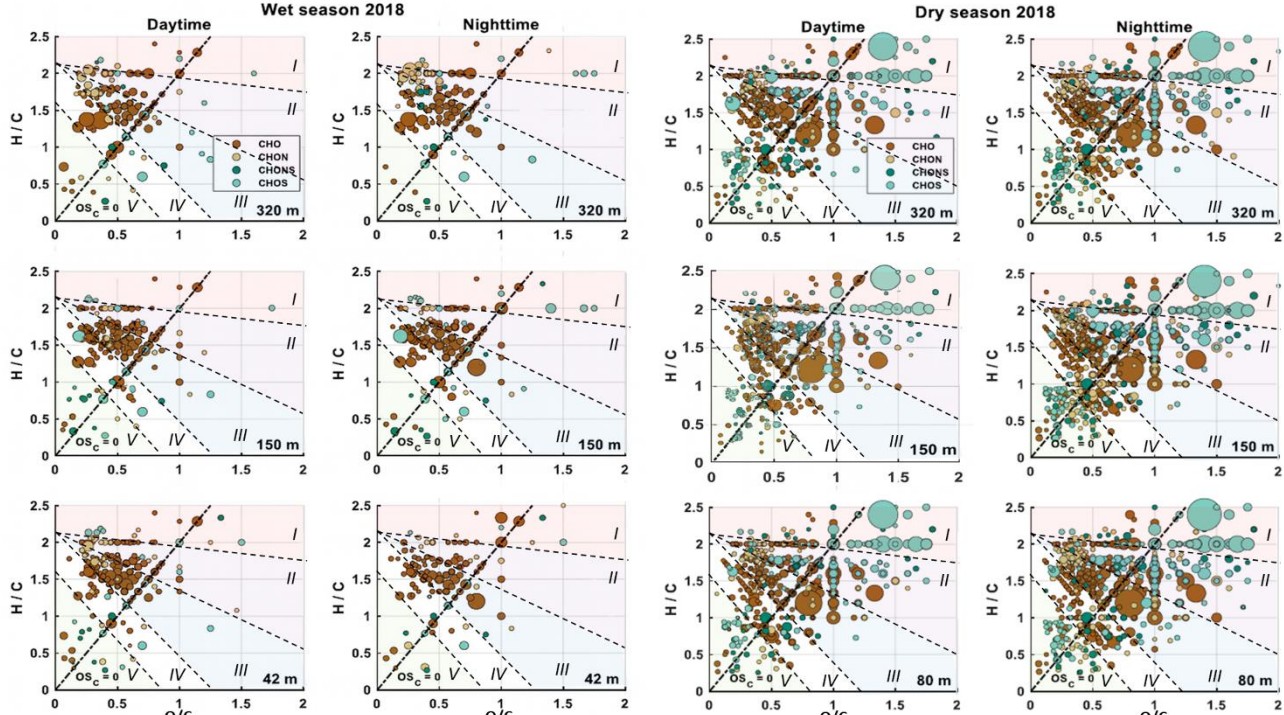

**Figure 3: Van Krevelen plots from the wet season 2018 (left panel) and dry season 2018 (right panel) during daytime and nighttime at 42 m, 150 m, 320 m height and 320 m height and dry season 2018 (right panel) during daytime and nighttime at 80 m, 150 m, and 320 m height. Included are only molecular formulae that were present in more than one of the samples, respectively. The background signals are subtracted. The size of the data points represents the signal intensity of the corresponding peak. The four subgroups CHO, CHON, CHONS and CHOS are distinguished with different colors. Compounds located on the black dashed line have an average carbon oxidation state of 0 ($OSC = 0$). The VK diagram was divided into five areas: I for very highly oxidized organic compounds, II for highly oxidized compounds, III for intermediately oxidized compounds, IV for oxidized unsaturated organic compounds and V for highly unsaturated organic compounds. The five areas were defined according to Zhang et al., 2021. Area (I) is mainly characterized by CHOS molecules with at least one sulfate functional group and C4 and C5 backbones. Area (II) is dominated by highly oxidized compounds representing aged OA and biomass burning products. Area (III) shows mainly aliphatic species. Area (IV) is dominated by unsaturated hydrocarbons with low oxygen content and aromatic structures while area (V) shows mainly combustion related species.**

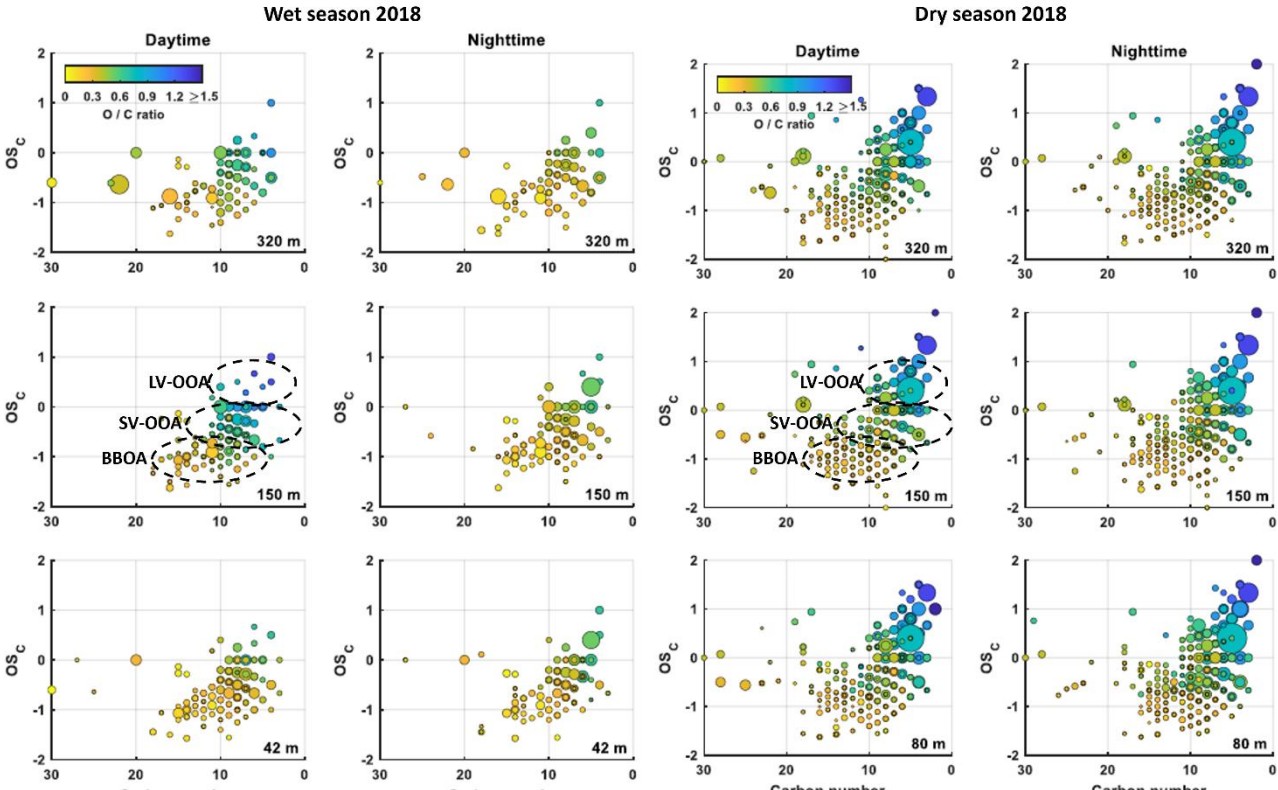

**Figure 4: Carbon oxidation state (OSC) plots for all detected CHO species during the wet season 2018 at daytime and night time at 42 m, 150 m, 320 m and dry season 2018 (right) at daytime and nighttime at 80 m, 150 m and 320 m sampling height. Included are only molecular formulae that were present in more than one of the samples, respectively. The background signals are subtracted. The size of the data points represents the signal intensity of the corresponding peak. The color code illustrates the degree of oxygenation. The black dashed areas are related to low- volatility (LV-OOA), semivolatile (SV-OOA) oxygenated organic aerosol and biomass burning organic aerosol (BBOA).**

### 3.3.2 Saturation vapor pressure

The saturation vapor pressure $C^0$ is an important thermodynamic property that regulates the gas-to-particle partitioning properties of organic molecules and enables their classification as VOCs, intermediate volatile organic compounds (IVOCs), semi-volatile OCs (SVOCs), low-volatility OCs (LVOCs) and extremely low-volatility OCs (ELVOCs) (Pankow, 1994; Odum et al., 1996; Murphy et al., 2014). The relationship between molecular weight, chemical composition and saturation vapor pressure has already been used to describe the chemical evolution of organic aerosols (Shiraiwa et al., 2014; Li et al., 2016). A case study for organic species observed in the dry and wet season 2018 is presented in Figure 5 showing 2-D maps of molecular weight versus saturation vapor pressure. Nearly all CHONS compounds are located within the LVOC/ELVOC range, and it is clear that many of the additional compounds detected in the dry season exhibit very low volatility, indicated by high molecular weights, and a more oxidized (aged) character relative to the wet season composition. Increased nitrogen oxide and $SO_2$ concentrations and thus the formation of N- and S-functionalized oxidation products, but also a lower removal by precipitation and thus longer residence times in the atmosphere could explain these differences between wet and dry season. These factors suggest that the observed compounds reflect a regional burning signature in the dry season. Several high molecular weight compounds with C8-C10 or C17-C20 backbones have also been identified as LVOC/ELVOC, suggesting oligomeric structures derived from isoprene and monoterpene precursors. It is hypothesized that these products are low volatile due to the incorporation of additional functional groups and the increase in carbon number in the molecular structure (Kroll and Seinfeld, 2008; Daumit et al., 2013). Two prominent ion signals at *m/z* 333.0859 ($C_{10}H_{22}O_{10}S$) and *m/z* 357.1557 ($C_{17}H_{26}O_8$) were previously reported in chamber studies with isoprene and α-pinene (Surratt et al., 2008; Kristensen et al., 2013). Most oligomers were detected in filter samples collected during the dry season, suggesting a stronger contribution of

the oligomerization reaction to SOA under these conditions. In addition to structures derived from isoprene and monoterpene precursors, compounds with molecular formulae such as $C_{14-15}H_{22-24}O_{3-7}$ were detected in the dry season 2018 (i.e. $C_{14}H_{22}O_7$;

$C_{15}H_{24}O_4$), contributing to the LVOC fraction. These signals are in line with oxidation products previously identified in chamber simulation experiments of β-caryophyllene ozonolysis (Gao et al., 2022) and could be assigned to β-caryophyllonic acid ($C_{15}H_{24}O_3$) as well as β-nocaryophyllonic acid and β-caryophyllinic acid ($C_{14}H_{22}O_4$) by comparison with authentic standard compounds (Table S5). The authors investigated the formation of SOA under varying nitrogen oxide levels at different temperatures. At a temperature of 313 K and the absence of nitrogen oxides, monomers (mainly $C_{14-15}H_{22-24}O_{3-7}$) and dimers

(mainly $C_{28-30}H_{44-48}O_{5-9}$) could be detected. In the presence of nitrogen oxides, which is more characteristic of the dry season conditions in 2018, most organonitrates were found as monomers with a C15 skeleton and one nitrate group ($C_{15}H_{23-25}O_{7-9}N$). However, in the wet and dry season 2018 no significant contribution of $C_{28-30}H_{44-48}O_{5-9}$ or $C_{15}H_{23-25}O_{7-9}N$ could be found. In case of CHON compounds with only one N atom, the dominant compound in the dry season 2018 was $C_{14}H_{27}NO_7$. This molecule has previously been identified as a product of toluene-derived SOA in chamber experiments (Zhang et al., 2020).

These findings imply that, in addition to biogenic precursors, aromatic anthropogenic emissions may also play a relevant role in shaping the low-volatility organic aerosol composition during the dry season.

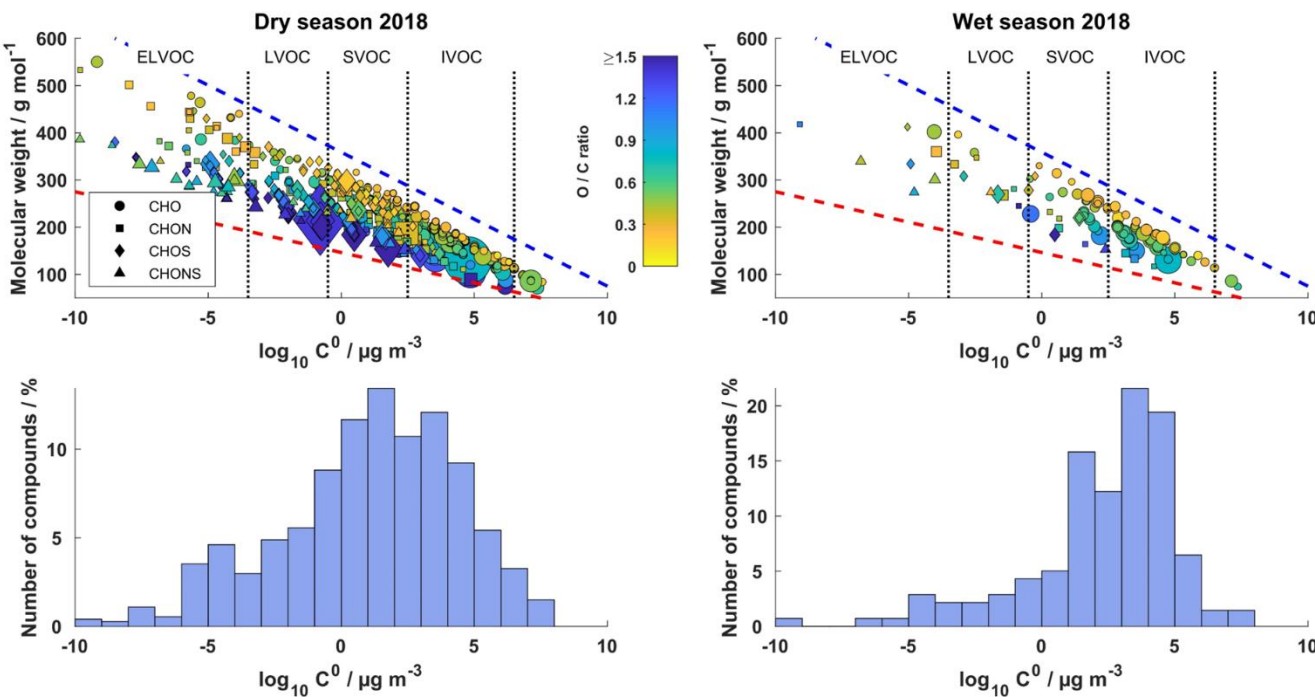

**Figure 5: Molecular classification of organic species for the dry season 2018 (left) and the wet season 2018 (right) according to their**
**saturation vapor pressure $C^0$. The color code describes the degree of oxygenation. The blue dashed line indicates linear alkanes $C_nH_{2n+2}$, while the red dashed line represents sugar alcohols $C_nH_{2n+2}O_n$ according to Li et al. (2016). The lower panel shows the distribution of all detected compounds among the respective classes.**

### 3.3.3 Kendrick mass analysis

Kendrick mass analysis is a valuable visualization tool for complex organic mixtures and was introduced by Kendrick (1963)
and later refined by Hughey et al. (2001). This technique has already been used to characterize organic aerosol samples (Lin et al., 2012; Rincón et al., 2012; Kourtchev et al., 2013). It enables the assignment of all signals of a homologous series when the elemental composition of a compound has been identified (Nizkorodov et al., 2011). Figure 6 compares the $KMD_{CH2}$ diagrams (see eq. 2.5 and 2.6) for the dry and wet seasons in 2018. Compounds differing only in their number of $CH_2$ units have the same KMD values. Homologous series can be identified as horizontal lines in KMD plots.

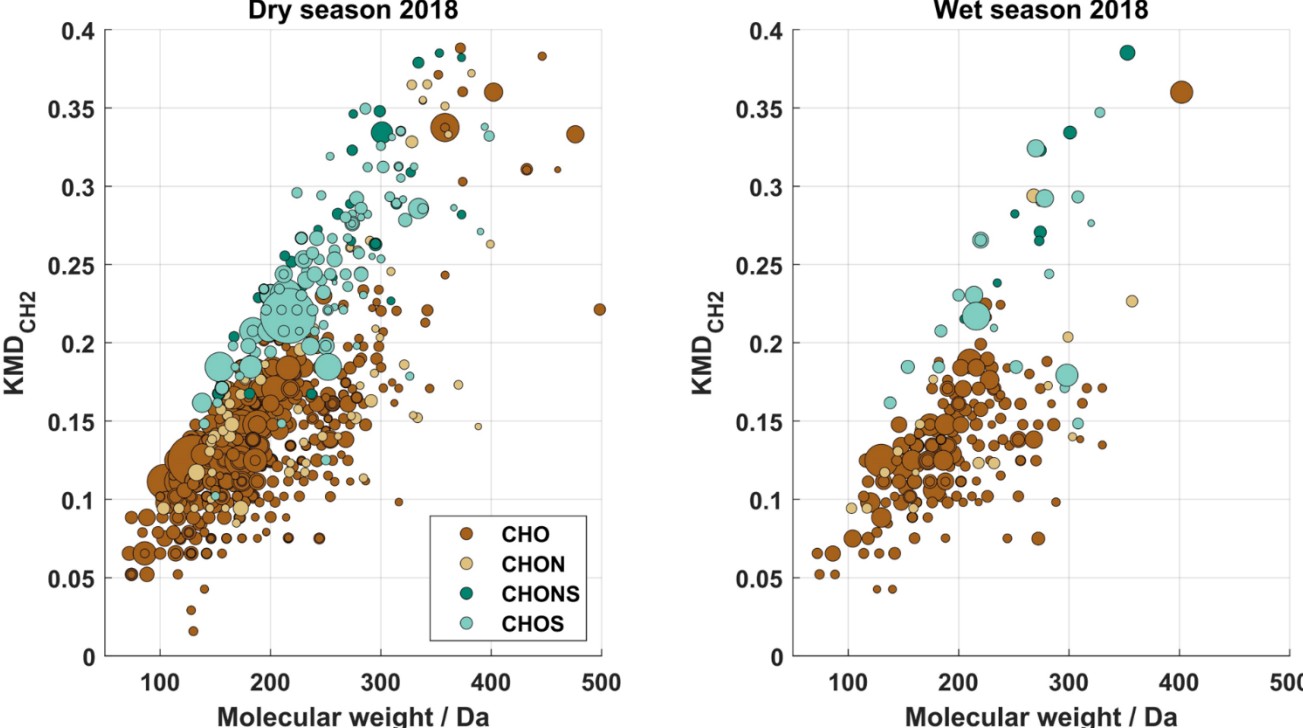


**Figure 6: CH₂-Kendrick diagrams for all signals detected at 150 m during the dry season 2018 (left) and the wet season 2018 (right), respectively. The size of the data points represents the signal intensity of the corresponding peak. The four subgroups are distinguished with different colors.**

As mentioned above, the dry season was characterized by biomass burning and the occurrence of aged organic aerosol

components, resulting in large-membered homologous series of CHO compounds with higher MW than in the wet season. Compounds with $KMD_{CH2} \leq 0.11$ appear to be unsaturated carboxylic acids with the general molecular formulae $C_nH_{2n}O_2$, $C_nH_{2n}O_3$, $C_nH_{2n-2}O_2$ and $C_nH_{2n-2}O_4$. The elemental composition suggests fatty acids, which have been described in OA-related studies as essential compounds of marine and terrestrial vegetation (Stephanou and Stratigakis, 1993; Rincón et al., 2012; Kourtchev et al., 2014b). Thus, long-range transport of marine aerosols or BBOA could explain the greater abundance of fatty

acids during the dry season (Oros and Simoneit, 2001; Tervahattu et al., 2002). In addition, a large number of CHOS compounds with $KMD_{CH2} \geq 0.21$ were detected exclusively during the dry season. Similar results were reported by Kourtchev et al. (2016), who found a larger number of homologous series of CHOS compounds in aerosol samples affected by anthropogenic emissions. These species are presumably highly oxygenated organosulfates, which are probably formed by heterogeneous reactions on acidic sulfate particles.

Another interesting ion was detected at *m/z* 186.1135 ($C_9H_{17}NO_3$) with high signal intensities, especially during the dry season 2018. This compound has already been observed in biomass burning chamber experiments and field measurements in the Amazon rainforest (Laskin et al., 2009; Kourtchev et al., 2016). In the present study, Kendrick mass analysis revealed evidence of a homologous series, which is shown in the KMD plot in Figure 7. The general molecular formula can be described as $C_nH_{2n-1}NO_3$, where n ranges from 3 to 17 depending on the filter sample. The elemental composition initially suggested a

nitrooxy functional group (-ONO₂), but MS[2] experiments (Figure S18) did not show the typical nitrate fragment. Instead, the observed fragments indicated the presence of a carbamate functional group (R-NH-COOH). Due to their zwitterionic structure, these species could be relevant for condensed phase chemistry in the atmosphere, similar to amino acids (Mopper and Zika, 1987; Milne and Zika, 1993; McGregor and Anastasio, 2001; Barbaro et al., 2011).

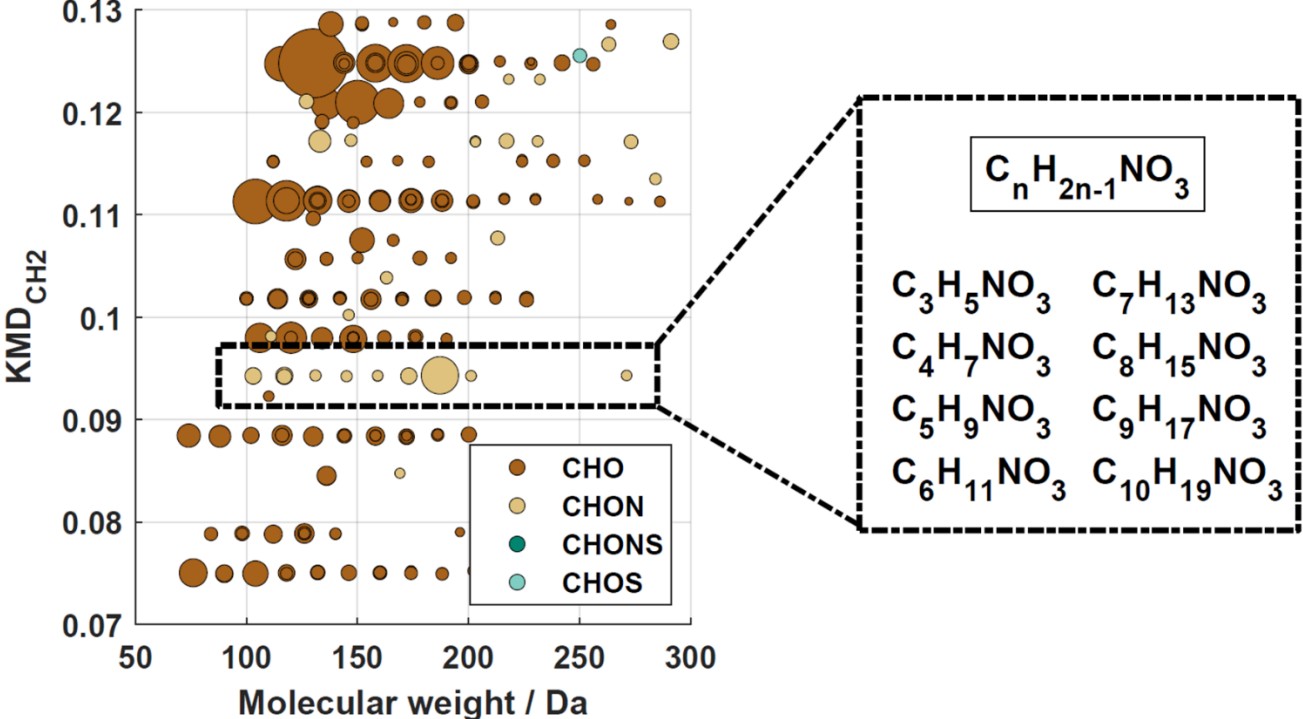


**Figure 7: Kendrick mass defect diagram for the dry season 2018. The size of the data points represents the signal intensity of the corresponding peak. The four subgroups are distinguished with different colors. The dashed box highlights a homologous series of CHON compounds with a molecular formula of $C_nH_{2n-1}NO_3$ with n = (3 – 17), potentially related to biomass burning emissions.**

## 4 Conclusion

The organic chemical composition of atmospheric OA particles from the Amazon rainforest was investigated at the ATTO station using high-resolution MS in combination with an UHPLC system. Between 2018 and 2019, a total of four measurement campaigns were carried out, including two dry and wet seasons, with one "clean" wet period, one wet period influenced by biomass burning and combustion activities and two "polluted" dry periods. This allows conclusions to be drawn about seasonal influences in a unique ecosystem that is characterized by the complex interaction of numerous emission sources. In this

analysis, a distinction was made between background compounds with a relatively low variability over time, and compounds with a much higher variability in their concentrations.

The results show a high fraction of isomeric structures in each data set, highlighting the need to use chromatographic separation methods for molecular identification of the aerosol components. Such an approach allows the reliable identification and quantification of a number of important marker compounds if reference compounds are available. However, the semi-

quantitative determination of other compound classes is also possible, particularly by reducing ion suppression. The majority of compounds measured in both seasons had molecular weights (MW) below 250 Da. In addition, the analysis of the dry season samples showed signals in the oligomeric range between 300 and 450 Da with lower intensities. In general, the background molecular composition of organic aerosols at ATTO showed a clear seasonality with large differences between the dry and wet seasons. A considerable number of suitable marker compounds for different SOA sources were identified. The most

important sources of background SOA at ATTO were the oxidation of isoprene and monoterpenes at both, high and low NOx concentrations and the formation of organosulfates. The results show that isoprene is the main precursor for S-containing SOA components at ATTO. The formation of isoprene OS probably occurs through the reactive uptake of IEPOX on acidic sulfate particles. It is assumed that in the wet season the sulfate particles reach the ATTO station mainly by long-range transport of marine aerosol, while in the dry season direct anthropogenic sources are also likely to be present. Furthermore, earlier studies

showed that organic sulfur emissions from soils increase with higher soil water content which indicates a stronger organosulfur soil source from the forest itself in the wet seasons. Isoprene organosulfates can have both, purely biogenic and anthropogenic

sources, whereby the biogenic sources appear to have a greater influence if the background OA composition is taken into account. This is further supported by the observation that the CHOS subgroup accounted for $(8 \pm 7)$ % in the wet season compared to only $(3 \pm 2)$ % in the dry season. Other dominant SOA groups that characterized the background were aromatic compounds whose source was most likely biomass burning. Thus, various oxygenated and nitrated hydrocarbons and oxidized polycyclic compounds showed significantly higher concentrations during the dry seasons. The evaluation of carbon oxidation states shows that LV-OOA and BBOA were the predominant and highly concentrated species in the dry seasons, while SV-OOA dominated in the wet seasons. We assume that a significant proportion of LV-OOA is formed by the aging and chemical processing of SOA components.

High signal intensities of nitroxyorganosulfates were observed in the dry seasons 2018 and 2019 as well as in the wet season 2019, whereby these were particularly derived from monoterpene precursors and correlated with the increased occurrence of fires and the $SO_2$ and NOx concentrations. Noteworthy are the low vapor pressures of the CHONS compounds, which therefore make a significant contribution to ELVOCs. In addition, we find evidence of oligomeric structures of biogenic SOA components, as well as dimers of sesquiterpene oxidation products, which could also contribute to the ELVOC fraction. At the same time, a large fraction of BBOA contributes to the total OA, as shown by the high Xc values. Consequently, unsaturated and aromatic compounds such as nitrocatechols and nitrophenols were mainly attributed to deforestation fires, which occurred mainly during the dry periods. Interestingly, the chemical composition of OA particles from the 2019 wet season revealed a similar OA composition as in the dry season, suggesting that combustion emissions played a larger role in the wet season 2019. Fossil fuel combustion in Europe or strong biomass burning in the Sahel savanna regions could be responsible via long-distance transportation across the Atlantic. Follow up studies to address these different long-range transport scenarios in dedicated case studies, providing a more detailed understanding of their respective contributions are in preparation.

## Data availability

The data sets are available at upon registration at https://www.attodata.org (last access: 25 January 2022), and more information can be given by Thorsten Hoffmann (t.hoffmann@uni-mainz.de).

## Author contributions

DL and SH contributed equally. DL, SH and TH planned and designed the study. DL conducted the aerosol sampling, the sample preparation and analysis, the evaluation and wrote the manuscript. SH conducted the method validation of sample preparation and analysis, the evaluation and wrote the manuscript. NZ helped to collect the aerosol samples and revised the manuscript. LK, BH, JW, CP, and SW provided the SMPS and trace gas data. The meteorological data was provided by MS. MCS and UP provided scientific input and revised the manuscript. All authors contributed to the discussion of the results as well as the finalization of the manuscript.

## Competing interests

The contact author has declared that none of the authors has any competing interests.

## Financial support

This research has been supported by the German Federal Ministry of Education and Research (BMBF, grant no. 01LK1602D)); the Brazilian Ministério da Ciência, Technologia e Inovação (MCTI/FINEP); the Amazon State University (UEA); Fundação de Amparo à Pesquisa do Estado do Amazonas (FAPEAM); LBA/INPA; and SDS/CEUC/RDS-Uatumã and the German Research Foundation (Deutsche Forschungsgemeinschaft DFG: HO 1748/19–1; HO 1748/23–1; TRR 301 – Project-ID 428312742).

This open-access publication was funded by Johannes Gutenberg University Mainz.


**Acknowledgments**

For the operation of the ATTO site, we acknowledge the support by the German Federal Ministry of Education and Research (BMBF Contract 01LK1602D) and the Brazilian Ministério da Ciência, Technologia e Inovação (MCTI/FINEP) as well as the
Amazon State University (UEA), FAPEAM, LBA/INPA, and SDS/CEUC/RDS-Uatumã. We also acknowledge the support of the Max Planck Society, and the Max Planck Graduate Center with the Johannes Gutenberg University Mainz (MPGC) and the Instituto Nacional de Pesquisas da Amazonia (INPA). Furthermore, we thank Florian Ditas and Maria Prass for help providing SMPS data. Particularly, we would like to thank the ATTO team members including Susan Trumbore, Alberto Quesada, Bruno Takeshi, Reiner Ditz, Stefan Wolff, Björn Nillius, Fernando Morais, Thomas Klimach, Roberta Pereira de
Souza, Jürgen Kesselmeier, Andrew Crozier, Sam Jones, Delano Campos, Juarez Viegas, Sipko Bulthuis, Francisco Alcinei Gomes da Silva, Isabella Diogenes, Hermes Braga Xavier, Nagib Alberto de Castro Souza, Antonio Huxley Melo Nascimento, Valmir Ferreira de Lima, Feliciano de Souza Coelho, André Luiz Matos, Wallace Rabelo Costa, Amauri Rodriguês Perreira, Adir Vasconcelos Brandão, Davirley Gomes Silva, Thomas Disper, Torsten Helmer, Steffen Schmidt, Uwe Schulz, Uwe Schultz, Karl Kübler, Olaf Kolle, Martin Hertel, Kerstin Hippler, Steffen Schmidt and all further colleagues involved in the
technical, logistical, and scientific support.

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
