# Peer review of "Comprehensive Non-targeted Molecular Characterization of Organic 2 Aerosols in the Amazon Rainforest"

_EGUsphere, 2025_

## Author Comment (AC1)

We would like to thank the reviewer for their valuable and constructive comments/suggestions that helped improve our manuscript. We have carefully addressed all suggestions and revised the manuscript accordingly. Below you will find our point-by-point responses. Reviewer comments and suggestions are written in black, responses in blue.

RC1: 'Comment on egusphere-2025-141', Anonymous Referee #1, 18 Mar 2025

The manuscript by Leppla et al. investigates the chemical composition and potential sources and chemistry of organic aerosols in two different wet and dry seasons in the Amazon rainforest with the deployment of an UHPLC-HR-Orbitrap mass spectrometer. It also compares the molecular composition and volatility of background compounds with relatively low variability and compounds with higher variability. The topic of this manuscript is very interesting. However, some revisions are needed before its possible publication on ACP. Please see my comments and questions below.

We appreciate the reviewer's assessment of the quality of the manuscript and thank for the work put into revising our manuscript.

**Major:**

More discussions on the comparison of the results at different heights are needed as the abstract underlined the height-resolved measurements. If the chemical composition is similar at different altitudes (Line 366-367, 390-392), then it seems not so important to include all heights in the main text (?). Is there any nocturnal differences in chemical composition below and above forest canopy? Could the authors comment on this?

We appreciate the reviewers's suggestion to provide a more detailed discussion of different heights. Due to the non-targeted approach in this study, it was challenging to identify significant height-resolved differences in the chemical composition of OA at ATTO, and the trends varied for particular compound classes, heights, seasons or daytime/nighttime samples. Nevertheless, we re-evaluated all height-resolved measurements and adjusted the focus to reflect the actual observations without aiming for specific trends.

As a result, we included a new chapter "3.2.3 Height-dependency of the background OA characteristics" in line 409ff for the height-resolved measurements of the background OA. For the discussion of the height-resolved variable OA characteristics at ATTO we revised Fig 3 and Fig 4 in the revised manuscript and Fig S13-S16 in the revised supplementary information, which display all measured heights now. Furthermore, we included new discussions about the observed differences in the chemical composition of OA at different heights in the lines 491ff; 532ff; 554ff; 556ff; 576ff, 590ff and 607ff (track-changes-version).

With regard to the nocturnal differences in chemical composition below and above the forest canopy only samples taken in the dry season 2019 can be considered since this was the only season when sampling was performed below canopy (0 m). The dry seasons are generally characterized by more uniform vertical distributions and enhanced mixing of aged, oxidized

species due to the prevalent southwesterly wind direction associated with higher surface roughness and increased turbulence. Therefore, height-resolved differences in the chemical composition in the dry seasons 2018 and 2019 are comparably low as convective processes lead to a more homogeneous chemical distribution. To better capture vertical gradients under low-mixing conditions, we implemented sampling below and above canopy during subsequent campaigns in the wet seasons for upcoming follow-up studies.

Nevertheless, we observed and discussed the following differences and included them in line 607ff in the track-changes version of the manuscript:

"However, the dry season 2019, which included sampling at three different times and at sub-canopy level (0 m), revealed unique patterns. SV-OOA and CHOS compounds in areas I and II peaked at 80 m during nighttime, consistent with gas-particle partitioning favored by cooler, stable layers above the canopy. CHON compounds were most abundant at 0 m and 80 m in the morning but diminished at 320 m. These patterns underline the importance of both vertical stratification and local chemical processing in modulating aerosol composition at different heights and times of day. Mendonça et al. (2025) noted that dry season nights at ATTO, characterized by southwesterly winds and enhanced surface roughness, can exhibit deeper but more turbulent boundary layers, allowing complex layering and submesoscale motions to form, which could explain the varied height-dependent signals observed in this campaign. Moreover, the presence of the highest CHOS and SV-OOA signals at 80 m during nighttime suggests a zone of active condensation and SOA formation just above the canopy, where gas-particle partitioning is favored by cooler and more stable stratification. The fact that CHON species were relatively suppressed at 320 m both during morning and daytime indicates limited upward transport of nitrogen-containing precursors or their rapid transformation near the surface. The chemical differences between 0 m and 80 m, particularly for CHON and CHO species in areas *II* and *III*, also suggest that in-canopy processes such as deposition, emissions, and light penetration significantly modulate chemical composition. Together, these observations highlight the sensitivity of nighttime aerosol chemistry to fine-scale vertical structure and suggest that sub-canopy and canopy-top levels may act as chemically distinct compartments in the nocturnal boundary layer during the dry season."

**Specific:**

Line 29-30. The reviewer didn't find the results in the main text on the forest canopy height being the main source of biogenic emissions or early terpene oxidation products. Wet season 2018 seems to have the filters collected at the closest height (40m) to the canopy height (35m); however, Wet season 2018 had less compounds (both background and variable ones) in Figure 2 and 3.

We thank the reviewer for pointing out the inconsistency. We agree that the canopy is not the main source of early terpene oxidation products at ATTO but rather a dominant one. Accordingly, we revised the sentence in line 30-31 (track-changes version) to:

"Height-resolved measurements showed biogenic emissions with higher concentrations of early terpene oxidation products at lower altitudes. "

Line 105-106. Could the authors explain the purpose/reason of sampling at different heights in wet and dry seasons (e.g. wet season 2018 at 42m and 150m but dry season at 0m and 80m)? How would this affect the result interpretation and comparison between different seasons? Could the authors comment on this?

The sampling heights selected during each campaign were influenced by a combination of exploratory objectives and logistical constraints. Due to a limited number of operational pumps (maximum three), only a subset of possible heights could be sampled in parallel. Initially, our design targeted three levels: 42 m (just above canopy), 150 m (boundary layer, mixing fractions), and 320 m (regional influences) during the wet season 2018. At the dry season 2018, logistical and spatial limitations made continued sampling at 42 m infeasible. Consequently, we switched to the 80 m platform, which provided a suitable alternative.
Following the wet season 2019, observations suggested limited vertical differentiation in OA composition due to enhanced mixing, so we redirected our focus to contrast the air masses below and above the canopy. Thus, the 150 m level was replaced with ground-level (0 m) sampling.

We have included the effects of the sampling heights for the result interpretation in the newly added sections on height-dependent OA composition.

Line 111. How many filters were collected at each height and each season? Please add this information.

We thank the reviewer for this suggestion. We have added the requested information to line 111ff (track-changes version):

"Ambient PM2.5 filter samples were collected at three different heights at the tower (wet season 2018: 42 m – 15 filters + 2 blank filters, 150 m – 14 filters + 2 blank filters, 320 m – 15 filters + 2 blank filters; dry season 2018 : 80 m - 18 filters + 2 blank filters, 150 m – 18 filters + 2 blank filters, 320 m – 18 filters + 2 blank filters; wet season 2019: 80 m – 22 filters + 2 blank filters, 150 m – 22 filters + 2 blank filters, 320 m – 22 filters + 2 blank filters ; dry season 2019: 0 m – 16 filters + 2 blank filters, 80 m – 14 filters + 2 blank filters, 320 m – 16 filters + 2 blank filters,) using borosilicate glass microfiber filter bonded with PTFE (Pallflex® Emfab, 70 mm diameter)."

Line 132-134. Why did the authors exclude fluorine in the data evaluation if recent studies have found fluorine-containing species in Amazon? Are they present in your dataset? What's their signal intensity contribution? Considering the high resolution of Orbitrap, it should be possible to identify fluorine-containing species, unless they are not present in this study.

We agree that the inclusion of fluorine-containing species could be valuable. However, in our case, processing the data with MZmine 2.30 produced unreliable results for fluorinated compounds. In step 7 of the data processing workflow (see supplement), molecular formulas were filtered based on isotope patterns within a 5 ppm tolerance. Fluorine has only one stable isotope ($^{19}F$) and a very low mass defect, which caused MZmine to assign false-positive formulas including F in cases where it was likely not present.
Therefore, we decided to exclude F-containing species from further analysis.
We hypothesize that these compounds constitute only a minor fraction of the total OA

composition at ATTO. Therefore, we are confident that our conclusions remain robust and representative, focusing on CHO, CHON, CHOS, and CHONS species.

Line 186-188. What did the authors mean in terms of "Only compounds that were observed in more than 75 % of all samples were defined as background compounds"? Is it based on the presence of the compound in the samples or based on some concentration criteria?

We thank the reviewer for pointing out the missing information. Only compounds that were observed in more than 75 % of all samples based on their presence were defined as background compounds. Signal intensities were not considered, as ESI-MS exhibits selective ionization efficiencies across compound classes. Therefore, intensity-based comparisons would be misleading. Due to the non-targeted nature of our approach only relative intensities of the same compound can be meaningfully compared. We have revised the sentence in line 195ff (track-changes version) as follows:

"Only compounds that were observed in more than 75 % of all samples based on their presence (identical molecular formula + retention time) were defined as background compounds. They presumably describe the local OA characteristics, as it is assumed that they are not dependent on individual emission events. For the variable OA characteristics, the evaluation was carried out by subtracting the peak areas of the previously determined background compounds from the peak areas of the total number of identified compounds (= background compounds + variable compounds). The remaining compounds are attributed to irregular atmospheric events, presumably caused by different meteorological conditions. Compounds that were only detected once in the respective data set were excluded as they were not considered representative."

Line 188-190. Are the remaining compounds unidentified compounds since the variable compounds are the remaining species of the total identified compounds from the background compounds? Would be nice to add total numbers of identified compounds for all groups (background compounds, variable compounds, and remaining compounds), and more importantly their signal intensity fraction.

We agree that the term "remaining compounds" could be confusing. We have clarified the text in line 199f (track-changes version) by replacing "remaining" with "variable" to avoid misunderstanding.

Regarding the signal intensity fractions ESI-MS is known for its highly compound-specific ionization efficiencies. Hence, signal intensity does not necessarily reflect actual concentration, but rather ionization potential.
A meaningful concentration comparison would require a targeted approach using calibration standards, a direction we are pursuing in future studies. Considering the scope of this work, we believe that the two tables (background and total compounds) already provide a comprehensive overview of OA molecular composition at ATTO.
To increase transparency, we added detailed explanations about calculating the relative contributions of CHO, CHON, CHONS and CHOS species in the captions of Table I and Table S2:

"Table I: Summary of all observed MS signals with unambiguous molecular formula assignment for the four measurement campaigns in 2018 and 2019 (wet season = WS, dry season = DS). The signals are divided into four subgroups regarding their elemental composition. The relative contribution of the subgroups was calculated by dividing the number of compounds of the particular subgroup by the total number of compounds. Additionally, the average values[1] of molecular weight (MW), carbon oxidation state ($OS_C$), aromaticity index (Xc), and isomeric fraction are listed."

"Table S12: Average values[1] of the detected background ions for the wet and dry seasons in 2018 and 2019. The listed molecules were detected in at least 75% of all corresponding samples. The signals are divided into four subgroups regarding their elemental composition. The relative contribution of the subgroups was calculated by dividing the number of compounds of the particular subgroup by the total number of compounds. Additionally, the average values[1] of molecular weight (MW), carbon oxidation state ($OS_C$) and aromaticity index (Xc) are listed."

Line 196-202. Please add the signal fractions of compounds with MW below 250 Da, 300-450 Da, and above 450 Da correspondingly. Also a typical mass spectra would be very informative.

We thank the reviewer for this suggestion. We have added the requested information in line 205ff in the revised manuscript:

"A total of 2336 molecular formulas could be assigned, of which 699 were in the range of < 250 Da, 1309 between 250 Da and 450 Da, and 328 above > 450 Da. Typical mass spectra are shown in Fig. S20."

Representative mass spectra have been added in the supplementary information in Figure S20.

Line 208-211. Would be nice to add the compound subgroup contributions from the mentioned remote/suburban/urban environments in the literature. Also the elemental ratios obtained in this study in Line 216 could be added.

We thank the reviewer for the suggestion and have included the subgroup contributions of CHON and CHONS (negative ionisation mode) from selected studies across remote, suburban, and urban environments. The revised sentence in line 218 (track-changes version) now reads:

"This trend is in good agreement with similar studies from remote environments (e.g. Amazonia, Brazil: $CHO^{(-)}$ with 58-63 %, $CHON^{(-)}$ with 25-30 %, $CHOS^{(-)}$ with 10 %, $CHONS^{(-)}$ with 2 %, Kourtchev et al., 2016; Hyytiälä, Finland: $CHO^{(-)}$ with 54.8 ± 2.2 %, $CHON^{(-)}$ with 21 ± 3 %, $CHOS^{(-)}$ with 16 ± 3 %, $CHONS^{(-)}$ with 5.4 ± 2.2 %, Kourtchev et al., 2013), while studies from a suburban and urban environment revealed enhanced contributions of CHON and CHONS compounds (Pearl River Delta region, China: $CHON^{(-)}$ with 34 %, $CHONS^{(-)}$ with 8 %, Lin et al., 2012; Cambridge, UK: $CHON^{(-)}$ with 33 %, $CHONS^{(-)}$ with 21-26 %, Rincón et al., 2012; Shanghai, China: $CHON^{(-)}$ with 21-23.7 %, $CHONS^{(-)}$ with 11.2-16.6 % , Wang et al., 2017), proving an increased relevance of nitrogen and sulfur chemistry in more polluted areas."

We also thank the reviewer for suggesting the inclusion of elemental ratios. The following sentence was added in line 228 in the revised manuscript:

"Comparable values were reported from the boreal forest in Hyytiälä, Finland (0.58 and 1.54 for O/C and H/C, respectively) (Kourtchev et al., 2013)."

Line 297. Which figure or table showed this? Please specify. Also considering important contributions of biogenic emissions during the wet season, could the authors explain the reason why the intensities of a-pinene oxidation products were higher in the dry than in the wet seasons?

We thank the reviewer for this valuable comment. The sentence was revised in line 312 (track-changes version) as follows:

"The wet season 2019 campaign was characterized by intense ion signals at $m/z$ 157.0506 ($C_7H_{10}O_4$) and $m/z$ 171.0662 ($C_8H_{12}O_4$) (Fig S9), attributed to limonene and α-pinene oxidation products, among others (Hammes et al., 2019; Eddingsaas et al., 2012; Thoma et al., 2022; Florou et al., 2024) suggesting that biogenic sources are dominant contributors to SOA loading at ATTO. Furthermore, the intensities of α-pinene oxidation products were higher in the dry than in the wet seasons. This can be explained by increased ambient temperatures (Fig S1; Fig S2) and photosynthetic active radiation facilitating the emission rates of monoterpenes (Guenther et al., 1991; Kesselmeier and Staudt, 1999). This also affects the diel variation with prevalent concentrations during the day."

Line 343-344. Why did 2019 dry season have fewer HOM? Is it related to the higher NO levels or more fires during this period?

We thank the reviewer for this interesting question.

Unfortunately, NOx data for the dry season 2019 are unavailable due to instrument failure (Fig. S8). Nevertheless, Pfannerstill et al. (2020) reported that although total OH reactivity at 80 m was slightly higher in September 2019 ($29.1 \pm 10.8$ $s^{-1}$) than in October 2018 ($28.1 \pm 7.9$ $s^{-1}$), this increase was not driven by monoterpenes (key HOM precursors) but rather by more oxidized VOCs such as aldehydes and organic acids.
This shift indicates a more aged, photochemically processed air mass, possibly linked to fire emissions. Although $NO_x$ were not measured directly, it is hypothesized that biomass burning contributed to a chemical regime where elevated NO levels may have suppressed $RO_2$ autoxidation, thereby inhibiting HOM formation. Together with lower monoterpene input, a combination of reduced precursor availability and a shift in chemical conditions likely explain the reduced HOM observations.

Pfannerstill, E. Y., Reijrink, N. G., Edtbauer, A., Ringsdorf, A., Zannoni, N., Araújo, A., Ditas, F., Holanda, B. A., Sá, M. O., Tsokankunku, A., Walter, D., Wolff, S., Lavrič, J. V., Pöhlker, C., Sörgel, M., and Williams, J.: Total OH reactivity over the Amazon rainforest: variability with temperature, wind, rain, altitude, time of day, season, and an overall budget closure, Atmos. Chem. Phys., 21, 6231–6256, https://doi.org/10.5194/acp-21-6231-2021, 2021.

We have strengthened our discussion regarding the different HOM availabilities accordingly in line 366ff (track-changes version).

Line 378-386. Would be nice to have a table for variable compounds similar to Table S2 (for background compounds) in SI. Also since the authors separated daytime vs nighttime for the variable compounds, why didn't you do the same for background compounds as well? Or is there no difference for day vs night for background compounds?

We thank the reviewer for these suggestions. The background compounds were not separated into day and night to simplify the processing pipeline. They were defined as compounds present in ≥75 % of all individual filters (day + night) per season. Mean signal intensities were then calculated per height for each season, allowing straightforward subtraction from the total OA signal without temporal segmentation. We agree that a day/night separation of background compounds would enhance detail but chose to avoid this due to data volume and focus. This will be addressed in future studies using targeted approaches, enabling precise comparisons with quantified concentrations.

Given the manuscript's length and the inclusion of new sections on height-dependence, we decided not to include an additional table for variable compounds. We feel this would not provide significant new information at this stage.

Line 393-395.

- If the authors would focus only on compounds with high intensities in the five areas, the reviewer would suggest to have a table for them similar to Table 2, and label them in Figure 3 similar to Figure 1.

We thank the reviewer for this suggestion. We agree that highlighting dominant peaks in areas *I–V* as in Table II and Figure 1 would be informative. However, due to the expansion of the discussion on vertical distribution, the figures now address broader trends and no longer focus solely on the most intense signals.
To maintain clarity, we retained the existing figure style but added new versions reflecting all height levels.

-Also $C_5H_{12}O_7S$ was already discussed in the background compound in Table 1. The reviewer would assume it should not be present in the variable compound group here as well. Same for $C_5H_6O_4$ in Line 427 which was also listed in Table 1.

We acknowledge the potential confusion and clarified the variable compound definition in line 197 (track-changes version):

"For the variable OA characteristics, the evaluation was carried out by subtracting the peak areas of the previously determined background compounds from the peak areas of the total number of identified compounds (= background compounds + variable compounds)."

Thus, a compound may appear in both categories if it also shows episodic enhancements. Additionally, different isomers may be present.

Line 459-463. Do you mean the dry season 2018 (in Line 459) had higher levels of CHON species at night than day? Is it a typo? It doesn't seem to be the case for wet season since there were very few CHON species both day and night.

We thank the reviewer for pointing out the inconsistiency. Indeed, it was a typo. The sentence in line 567f (track-changes version) has been corrected as follows:

"However, the dry season of 2018 shows a higher presence of CHON species during nighttime compared to daytime samples."

Line 468-473. What's the dominant species for these combustion-related highly unsaturated organic compounds?

We thank the reviewer for this insightful question. In our dataset, no single compound or small group consistently dominated the class of combustion-related, highly unsaturated organic species across all seasons. Their composition varied notably, likely reflecting different combustion sources, degrees of atmospheric aging, and air mass origins. For this reason, we refrained from highlighting specific compounds to avoid overinterpreting seasonally limited observations. We now clarify this point in line 584 of the manuscript (track-changes version):

"No single dominant species was identified among the combustion-related highly unsaturated organic compounds, as their composition varied substantially between seasons."

Line 520-525. Would be nice to have similar plots for the year 2019 data in SI. Also for background compounds from section 3.2.

We appreciate the reviewer's suggestion to add corresponding plots for the 2019 datasets. While we acknowledge that this could offer additional context, we deliberately focused on two representative seasons to clearly demonstrate the major seasonal contrasts in OA composition. This decision was taken to maintain a manageable manuscript length and analytical clarity. We believe that the presented datasets already capture the key features relevant to our conclusions.

We hope the reviewer agrees that this focused selection supports a clear and accessible presentation of the results.

Line 526-529. The authors know the molecular formulae of the compounds with MW between 500-600, and therefore it's possible compare the dominant species to the sesquiterpene oxidation products from Gao et al., 2022.

We thank the reviewer for this important suggestion. We have re-evaluated the detected molecular formulas in the 500–600 Da range. Two key species were identified: $C_{21}H_{21}NO_{14}$ (MW 533.0026 Da) and $C_{28}H_{22}O_{12}$ (MW 550.1105 Da). Both exhibit high aromaticity indices ($Xc$ = 2.6 and 2.83, respectively), suggesting they are likely aromatic species. We hypothesize that they may be derived from lignin-like biomass burning products.

Nevertheless, we detected sesquiterpene oxidation product signatures (e.g., $C_{14}H_{22}O_7$, $C_{15}H_{24}O_4$), which are in agreement with the findings of Gao et al. (2022). However, dimer species ($C_{28-30}H_{44-48}O_{5-9}$) derived from sesquiterpene oxidation products and dominant organonitrate compounds ($C_{15}H_{23-25}O_{7-9}N$) identified in their chamber studies did not appear significantly in our samples. We have strengthened the discussion in terms of sesquiterpene oxidation product monomers in line 667 of the track changes version as follows:

"Most oligomers were detected in filter samples collected during the dry season, suggesting a stronger contribution of the oligomerization reaction to SOA under these conditions. In addition to structures derived from isoprene and monoterpene precursors, compounds with molecular formulae such as $C_{14-15}H_{22-24}O_{3-7}$ were detected in the dry season 2018 (i.e. $C_{14}H_{22}O_7$; $C_{15}H_{24}O_4$), contributing to the LVOC fraction. These signals are in line with oxidation products previously identified in chamber simulation experiments of β-caryophyllene ozonolysis (Gao et al., 2022) and could be assigned to β-caryophyllonic acid ($C_{15}H_{24}O_3$) as well as β-nocaryophyllonic acid and β-caryophyllinic acid ($C_{14}H_{22}O_4$) by comparison with authentic standard compounds (Table S5). The authors investigated the formation of SOA under varying nitrogen oxide levels at different temperatures. At a temperature of 313 K and the absence of nitrogen oxides, monomers (mainly $C_{14-15}H_{22-24}O_{3-7}$) and dimers (mainly $C_{28-30}H_{44-48}O_{5-9}$) could be detected. In the presence of nitrogen oxides, which is more characteristic of the dry season conditions in 2018, most organonitrates were found as monomers with a C15 skeleton and one nitrate group ($C_{15}H_{23-25}O_{7-9}N$). However, in the wet and dry season 2018 no significant contribution of $C_{28-30}H_{44-48}O_{5-9}$ or $C_{15}H_{23-25}O_{7-9}N$ could be found. In case of CHON compounds with only one N atom, the dominant compound in the dry season 2018 was $C_{14}H_{27}NO_7$. This molecule has previously been identified as a product of toluene-derived SOA in chamber experiments (Zhang et al., 2020). These findings imply that, in addition to biogenic precursors, aromatic anthropogenic emissions may also play a relevant role in shaping the low-volatility organic aerosol composition during the dry season."

Line 547-548. Would be nice to have similar plots for the year 2019 data in SI as those in Figure 6 and 7. Also for background compounds from section 3.2.

We thank the reviewer for this constructive idea. However, to maintain a concise manuscript with a focused narrative, we chose to limit such visualizations to the most illustrative periods (wet and dry season 2018). Expanding the figure set to include 2019 would, in our view, dilute the core messages without providing substantial additional insights.

We plan to include such extended data in future follow-up work on interannual variability.

Line 581. Based on the discussions e.g. in Line 480-481 that the wet season 2019 was significantly influenced by biomass burning and combustion activities (Figure S11), the reviewer is wondering whether the wet season 2019 can still be classified as "clean" periods. Also the 7-d HYSPLIT backward trajectories in Figure S3 also shows the wet season 2019 had contact/source from the African continent but not the case for the wet season 2018.

We agree with the reviewer that "clean" is not an appropriate descriptor for wet season 2019. We have updated the sentence in line 725f (track-changes version) to:

"Between 2018 and 2019, a total of four measurement campaigns were carried out, including two dry and wet seasons, with one "clean" wet period, one wet period influenced by biomass burning and combustion activities and two "polluted" dry periods.".

Figure 5. Please change the y axis of lower panels from contribution of number of compounds" to "contribution of signal intensity" (Volatility Basis Set VBS; Donahue et al., 2006), since histogram of number of compounds doesn't equal to their role in volatility.

We thank the reviewer for this helpful suggestion. However, since ESI-MS has compound-specific ionization efficiencies, signal intensities may not reflect actual concentration contributions. To avoid misleading interpretations, we chose to calculate volatility distributions based on the number of detected compounds rather than signal intensities.

We agree that signal-based VBS classification is desirable, but such quantification would require a calibrated, targeted approach with internal standards, which is outside the scope of the current non-targeted study.

**Technical:**

Line 46. Would change "their nucleation" to "their oxidation products' nucleation".

We implemented the change in line 45ff (track-changes version) as follows:

"While primary organic aerosols (POA) are emitted directly into the atmosphere (e.g., combustion of biomass), secondary organic aerosols (SOA) are produced by oxidative processes and transformations of volatile organic compounds (VOC), followed by their oxidation products' nucleation and/or condensation in the atmosphere (Seinfeld and Pankow, 2003)."

Line 83-84. Seems repetition with the sentences in the previous paragraph.

We reorganized the paragraph as follows in line 80ff (track-changes version):

"In this study, LC-HR-MS was applied to analyze aerosol filter samples collected at the ATTO site during multiple campaigns across 2018 and 2019. Organic aerosols were sampled at distinct heights (0 m, 42 m, 80 m, 150 m, and 320 m) to investigate vertical gradients in chemical composition. This height-resolved sampling strategy was designed to capture the influence of biogenic emissions near the forest canopy, photochemical transformation processes above the canopy, and the impact of regionally transported, aged aerosol in the upper boundary layer. In particular, sub-canopy and near-canopy samples reflect local sources and early secondary organic aerosol formation, whereas the 320 m platform provides insight into regional and long-range transport contributions. These data enable a more detailed separation of local and regional influences on OA composition. It is important to note, however, that vertical differences are strongly influenced by seasonal and diurnal meteorological variability, including boundary layer dynamics, atmospheric mixing, and air mass history. Thus, height-dependent measurements must be interpreted within the broader context of meteorological conditions. Non-targeted high-resolution analytical techniques were employed to characterize the molecular composition of OA in relation to emission sources, (trans)formation processes, and atmospheric conditions. This molecular-level understanding contributes to improved representation of aerosol–cloud–climate interactions in atmospheric chemistry and climate models, particularly for pristine yet rapidly changing tropical regions such as the Amazon."

Line 84-86. Also seem a bit repetition with the sentences in line 80-81. Please consider combine them.

See above.

Line 201. Change "ions" to "molecules with MW".

We implemented the change in line 209ff (track-changes version) as follows:

"While several studies have shown that ozonolysis of biogenic VOCs (e.g. α-pinene, β-pinene, isoprene) in smog chamber experiments produces compounds with high molecular weight and increased signal intensities in the oligomeric range (Kourtchev et al., 2014a; Reinhardt et al., 2007), molecules with MW above 450 Da contributed insignificantly to the total number of compounds in the present filter samples."

Line 236. Change "particle phase" to "particles".

We implemented the change in line 248f (track-changes version) as follows:

"Higher wind speeds of up to $15 - 20$ m s$^{-1}$ were observed during the wet seasons (dry seaons $10-15$m s$^{-1}$), leading to a dilution of the particles at ATTO (Figure S 3)."

Line 262-322. Please consider removing the bold headlines (e.g. "General", "Isoprene SOA") in each of the paragraphs. Also the case for the bold headlines in section 3.3.1 for the five classes.

We changed the bold headlines in all the paragraphs.

Line 336. Change "nuclei" to "clusters".

We implemented the change in line 356ff in the track-changes version as follows:

"For example, highly oxidized molecules (HOM) with sufficiently low volatility can form clusters, which leads to the formation of new particles and subsequent particle growth, even of small particles (Bianchi et al., 2016; Molteni et al., 2016)."

Line 358. Typo for LV-OOA. Same for Line 452: typo for SV-OOA. Also Figure 4.

Thank you for pointing this out. We corrected the typos in line 392, 552 and Fig 4 in the revised manuscript accordingly.

Line 477-478. Seems repetition with the previous sentence.

Thank you for pointing this out. We rearranged the whole paragraph in line 593 (track-changes version) as follows:

"The wet season of 2019 stands out due to individual pollution events that significantly influenced aerosol composition at the ATTO site. The number of components and their intensity increase across areas *I–V* for the wet season 2019 (Fig. S13), compared to the wet season 2018 (Fig. 3) and the dry season of the same year (Fig. S14). The wet season 2019 also shows higher levels of CHON and CHO species relative to the 2019 dry season. The concentrations of OS related to isoprene remain comparable to those observed during the dry seasons of 2019 and 2018. In particular, area *V* in Fig S13 and the BBOA area in Fig S15 highlight the impact of biomass burning during this period. This is further supported by elevated BC concentrations (Fig S17). While the background OA characterization still shows typical wet season features, such as low BBOA contributions, a dominance of early-stage monoterpene oxidation products, minimal aging effects (Fig. S12), and absent signals in area *V* (Fig. S9), it becomes

evident that, with the inclusion of individual events, episodes of considerable pollution were present, temporarily altering the aerosol profile. These deviations likely reflect enhanced fire or combustion activity upwind or near the ATTO site during this period (Table S3). Additionally, emissions from biomass burning in the Sahel savanna regions, transported across the Atlantic Ocean (Fig S5), could have contributed to the observed aerosol characteristics (Holanda et al., 2020; Holanda et al., 2023)."

Line 599. Change "what" to "which".

We changes "what" to "which" in line 745 in the track-changes version.

Table 1 and S2. Change "Signals" to e.g. "Number of compounds detected".

We changed "signals" into "number of compounds" in Table 1 and S2 accordingly.

**Reference:**

Donahue, N. M., Robinson, A. L., Stanier, C. O., and Pandis, S. N.: Coupled partitioning, dilution, and chemical aging of semivolatile organics, Environ. Sci. Technol., 40, 2635–2643, 2006.

**Citation**: https://doi.org/10.5194/egusphere-2025-141-RC1

Author comment:

We hope that the revised manuscript and the responses provided here fully address the reviewer's concerns. We would be happy to clarify any remaining points.

---

## Author Comment (AC2)

We would like to thank the reviewer for their valuable and constructive comments/suggestions that helped improve our manuscript. We have carefully addressed all suggestions and revised the manuscript accordingly. Below you will find our point-by-point responses. Reviewer comments and suggestions are written in black, responses in blue.

RC2: 'Comment on egusphere-2025-141', Anonymous Referee #2, 06 May 2025

This study investigated the molecular-level chemical characteristics of organic aerosols collected from the Amazon rainforest during several seasons in 2018 and 2019. Orbitrap MS data provided extensive information on organic molecules, offering valuable insights for readers in this field. I recommend that the authors address the following comments:

We thank the reviewer for the constructive feedback and the effort dedicated to improving our manuscript.

Main Evaluation:

Although the authors emphasize the use of vertical resolution sampling in the abstract, the main text lacks sufficient comparison and discussion of results at different sampling heights. As shown in lines 366–367 and 390–392, the chemical compositions at different altitudes appear to be largely similar. If this is indeed the case, it is necessary to further explain the rationale for sampling at multiple heights. Currently, the results do not fully demonstrate the scientific value of height-resolved measurements.

We thank the reviewer for the suggestion to provide a more detailed discussion of the height-resolved measurements. Due to the non-targeted approach applied in this study, it was challenging to identify consistent vertical trends in the chemical composition of OA at ATTO, as the results varied across compound classes, height levels, seasons, and between day- and nighttime samples. Nevertheless, we re-evaluated all height-resolved data and adjusted the discussion to reflect the actual observations, without aiming to generalize specific trends.

As a result, we added a new section ("3.2.3 Height-dependency of the background OA characteristics") in line 409ff in the revised manuscript focusing on the vertical distribution of background OA. In addition, Figures 3 and 4 in the main manuscript and Figures S13–S16 in the supplementary information have been revised to display all measured heights. New discussions on the observed vertical differences in OA composition were included in lines 491ff; 532ff; 554ff; 556ff; 576ff, 590ff and 607ff of the revised manuscript.

The authors are advised to strengthen the discussion in the following areas:

1. Are there observable chemical differences between sampling heights under nighttime or specific meteorological conditions? For example, do chemical components above and below the forest canopy exhibit nighttime differences due to aerosol deposition, vertical mixing, or local chemical reactions?

We thank the reviewer for these insightful questions that we also consider to be relevant for our discussion. With regard to nocturnal differences in chemical composition below and above the forest canopy, only samples from the dry season 2019 can be considered, as this was the only period when sub-canopy sampling at 0 m was conducted. Dry season conditions are generally associated with more uniform vertical distributions and enhanced mixing of aged, oxidized compounds, driven by prevailing southwesterly winds, increased surface roughness, and turbulent conditions. Consequently, height-resolved differences in OA composition during the dry seasons of 2018 and 2019 were generally low, due to convective mixing leading to more homogeneous chemical distributions. To better capture vertical gradients under more stable conditions, subsequent wet season campaigns for follow-up studies included sampling both, below and above the canopy.

Nevertheless, notable differences were observed in the dry season 2019 and are discussed in line 607ff of the revised manuscript:

"However, the dry season 2019, which included sampling at three different times and at sub-canopy level (0 m), revealed unique patterns. SV-OOA and CHOS compounds in areas *I* and *II* peaked at 80 m during nighttime, consistent with gas-particle partitioning favored by cooler, stable layers above the canopy. CHON compounds were most abundant at 0 m and 80 m in the morning but diminished at 320 m. These patterns underline the importance of both vertical stratification and local chemical processing in modulating aerosol composition at different heights and times of day. Mendonça et al. (2025) noted that dry season nights at ATTO, characterized by southwesterly winds and enhanced surface roughness, can exhibit deeper but more turbulent boundary layers, allowing complex layering and submesoscale motions to form, which could explain the varied height-dependent signals observed in this campaign. Moreover, the presence of the highest CHOS and SV-OOA signals at 80 m during nighttime suggests a zone of active condensation and SOA formation just above the canopy, where gas-particle partitioning is favored by cooler and more stable stratification. The fact that CHON species were relatively suppressed at 320 m both during morning and daytime indicates limited upward transport of nitrogen-containing precursors or their rapid transformation near the surface. The chemical differences between 0 m and 80 m, particularly for CHON and CHO species in areas *II* and *III*, also suggest that in-canopy processes such as deposition, emissions, and light penetration significantly modulate chemical composition. Together, these observations highlight the sensitivity of nighttime aerosol chemistry to fine-scale vertical structure and suggest that sub-canopy and canopy-top levels may act as chemically distinct compartments in the nocturnal boundary layer during the dry season."

2. If overall differences are minimal, does this indicate strong vertical mixing or a stable boundary layer structure in the study area? Discussion of this would help explain the observed vertical uniformity.

We agree with the reviewer that the discussion would be improved by strengthening it regarding to boundary layer dynamics. A recent study of Mendonca et al., 2025 showed that the boundary layer dynamics at ATTO can vary significantly depending on the seasonal conditions. If prevalent southeasterly winds direction are prevalent, what is typical for dry seasons at ATTO (Fig S4), higher surface roughness and increased turbulence can be observed, leading to more

uniform vertical distributions and mixing of aged, oxidized OA. Contrarily, under wet season conditions, when the wind predominantly arrives from the northeast (Fig S5) and the surface roughness is relatively low, the nocturnal boundary layer is shallow (typically 80–120 m), limiting vertical exchange and favoring accumulation of semivolatile species near the canopy top and allowing upper levels (e.g. 320 m) to be decoupled and enriched in aged or transported aerosol components.

We included these points in our discussion in the revised manuscript in lines 494ff, 557ff, 608ff.

3. Does height-resolved sampling still provide value in identifying potential source regions, reaction mechanisms, or deposition processes? The authors are encouraged to further discuss this issue in their results.

We agree with the reviewer that pointing out the value of height-resolved sampling for identifying potential source regions, reaction mechanisms, or deposition processes would improve the discussion. We consider the newly included height-resolved interpretations of the chemical composition of OA in the revised manuscript as very insightful regarding the topics mentioned.

The revised manuscript underlines potential source regions in lines 495ff, 537f, 555f, 577ff. Furthermore, the influence of reaction mechanisms is pointed in lines 534f, 557, 611f and deposition processes are taken into account in lines 590f, 615f.

Specific comments:

Line 80: Although the study includes sampling at multiple heights, the introduction does not clearly articulate the scientific rationale or objectives of this vertical sampling design. Given that atmospheric composition, photochemical processes, and pollutant transformations vary significantly with altitude, it is essential to clarify why different altitudes were chosen and how this contributes to the overall research objectives.

We thank the reviewer for this valuable comment. The rationale for the vertical sampling design has now been clarified in the introduction section (lines 82ff in the revised manuscript) as follows:

"This height-resolved sampling strategy was designed to capture the influence of biogenic emissions near the forest canopy, photochemical transformation processes above the canopy, and the impact of regionally transported, aged aerosol in the upper boundary layer. In particular, sub-canopy and near-canopy samples reflect local sources and early secondary organic aerosol formation, whereas the 320 m platform provides insight into regional and long-range transport contributions. These data enable a more detailed separation of local and regional influences on OA composition. It is important to note, however, that vertical differences are strongly influenced by seasonal and diurnal meteorological variability, including boundary layer dynamics, atmospheric mixing, and air mass history. Thus, height-dependent measurements must be interpreted within the broader context of meteorological conditions."

Lines 105–110: How many filters were collected at each altitude for each season? Please add this information.

We thank the reviewer for this suggestion. We have added the requested information to line 111 (track-changes version):

"Ambient PM2.5 filter samples were collected at three different heights at the tower (wet season 2018: 42 m – 15 filters + 2 blank filters, 150 m – 14 filters + 2 blank filters, 320 m – 15 filters + 2 blank filters; dry season 2018 : 80 m - 18 filters + 2 blank filters, 150 m – 18 filters + 2 blank filters, 320 m – 18 filters + 2 blank filters; wet season 2019: 80 m – 22 filters + 2 blank filters, 150 m – 22 filters + 2 blank filters, 320 m – 22 filters + 2 blank filters ; dry season 2019: 0 m – 16 filters + 2 blank filters, 80 m – 14 filters + 2 blank filters, 320 m – 16 filters + 2 blank filters,) using borosilicate glass microfiber filter bonded with PTFE (Pallflex® Emfab, 70 mm diameter)."

Lines 185–186: The manuscript defines "background compounds" as those observed in over 75% of the samples. However, it is unclear how this classification was determined. Is it based solely on matching molecular formulas, or does it also consider MS² data to confirm structural similarity? Since compounds with the same molecular formula may have different structures and properties, clarifying the criteria used for this definition is crucial. I recommend that the authors clearly describe this aspect in the Methods section to ensure transparency and reproducibility.

We thank the reviewer for pointing out the missing information. Background compounds were defined as those detected in more than 75 % of all samples, based on their presence rather than signal intensity. As the analytical setup included an LC system upstream of the ESI-HR-Orbitrap-MS, molecular formula assignment was supplemented by retention time information. This additional dimension reflects compound polarity and allows for improved differentiation of chemical structures in cases of identical molecular formulas, and even enables the distinction between isomers.

We have revised the sentence in line 195 (track-changes version) as follows:

"Only compounds that were observed in more than 75 % of all samples based on their presence (identical molecular formula + retention time) were defined as background compounds. They presumably describe the local OA characteristics, as it is assumed that they are not dependent on individual emission events. For the variable OA characteristics, the evaluation was carried out by subtracting the peak areas of the previously determined background compounds from the peak areas of the total number of identified compounds (= background compounds + variable compounds). The remaining compounds are attributed to irregular atmospheric events, presumably caused by different meteorological conditions. Compounds that were only detected once in the respective data set were excluded as they were not considered representative."

Lines 196–198: Is this description a summary? What figure or table does the conclusion refer to?

We thank the reviewer for pointing out the lack of information. We changed the paragraph in the track-changes version in line 204 ff as follows:

"The chemical composition of the SOA samples was mainly influenced by seasonal effects during the measurement campaigns. A total of 2336 molecular formulas could be assigned, of which 699 were in the range of < 250 Da, 1309 between 250 Da and 450 Da, and 328 above > 450 Da. Typical mass spectra are shown in Fig. S 20. Molecular weights (MW) in the range below 250 Da could be assigned to the majority of observed substances in both seasons, while the aerosol composition in the dry season additionally showed signals in the oligomeric range between 300 and 450 Da with lower intensity."

Furthermore, a typical mass spectra were included in Fig S20.

Lines 208–210: It would be preferable to include specific contributions from composite subpopulations from remote/suburban/urban environments in the literature.

We agree with the reviewer and have included the subgroup contributions of CHON and CHONS (negative ion mode) from selected studies across remote, suburban, and urban environments. The revised sentence in line 218 (track-changes version) now reads:

"This trend is in good agreement with similar studies from remote environments (e.g. Amazonia, Brazil: CHO$^{(-)}$ with 58-63 %, CHON$^{(-)}$ with 25-30 %, CHOS$^{(-)}$ with 10 %, CHONS$^{(-)}$ with 2 %, Kourtchev et al., 2016; Hyytiälä, Finland: CHO$^{(-)}$ with 54.8 ± 2.2 %, CHON$^{(-)}$ with 21 ± 3 %, CHOS$^{(-)}$ with 16 ± 3 %, CHONS$^{(-)}$ with 5.4 ± 2.2 %, Kourtchev et al., 2013), while studies from a suburban and urban environment revealed enhanced contributions of CHON and CHONS compounds (Pearl River Delta region, China: CHON$^{(-)}$ with 34 %, CHONS$^{(-)}$ with 8 %, Lin et al., 2012; Cambridge, UK: CHON$^{(-)}$ with 33 %, CHONS$^{(-)}$ with 21-26 %, Rincón et al., 2012; Shanghai, China: CHON$^{(-)}$ with 21-23.7 %, CHONS$^{(-)}$ with 11.2-16.6 % , Wang et al., 2017), proving an increased relevance of nitrogen and sulfur chemistry in more polluted areas."

Lines 246–247: Given that CHONS account for only 1%, classifying them as major components seems questionable.

We agree with the reviewer and revised the sentence in line 257ff of the track-changes version as follows:

"The molecular formulae in the wet seasons are predominated by the CHO subgroup (90 ± 7) %, followed by CHOS with (8 ± 7) %, CHONS (1 ± 1) %, and CHON (1 ± 2) %. The dry seasons show a comparable predominance of the CHO subgroup (93 ± 3) % and an equal contribution of CHONS compounds (1 ± 1) % but an increased fraction of CHON compounds (4 ± 1) % and a decreased fraction of CHOS compounds (3 ± 2) %."

Lines 275–280: The manuscript discusses the effects of different NOx conditions on reaction mechanisms or product distributions but does not clearly define the criteria used to distinguish between high NOx and low NOx conditions. Additionally, the actual NOx concentrations observed in this study are not specified in the main text. I recommend that the authors clearly specify the classification criteria and provide the NOx data used in this work to enhance the scientific rigor and reproducibility of the discussion.

We thank the reviewer for this valuable suggestion. We agree that providing a clearer definition of the NO$_x$ regimes and including actual concentration data enhances the transparency and reproducibility of our discussion.

In the revised manuscript, we now explicitly define the classification of low-NO$_x$ and high-NO$_x$ conditions based on observed NO and NO$_2$ mixing ratios during the different campaigns in line 290 as follows:

"As shown in Figure S8, NO and NO$_2$ mixing ratios were generally <0.5 ppb and <1 ppb at 73.3 m and <2 ppb and <0.5 ppb at 0.05 m for the dry season 2018. For the wet seasons 2018 and 2019 NO mixing ratios were <0.25 ppb at 79.3 m and <1.5 ppb at 0.05 m whereas NO$_2$ mixing ratios were <0.05 ppb at 79.3 m and <0.1 ppb at 0.05 m. NO$_x$ data for the dry season 2019 are lacking due to instrument issues. These conditions are consistent with what is commonly defined as low-NO$_x$ regimes in previous chamber studies (e.g., Krechmer et al., 2015; Paulot et al., 2009; Nagori et al., 2019)."

Lines 409–411: Isoprene-derived organic sulfates (OS) are typically formed through photochemical oxidation, resulting in higher concentrations during the day than at night. However, in this study, the observed CHOS compounds showed higher abundances at night. I recommend that the authors more clearly emphasize the primary formation pathways of isoprene OS to strengthen the underlying principles and credibility of the conclusion that nighttime concentrations exceed daytime levels.

We thank the reviewer for this valuable comment and the opportunity to clarify our interpretation. It is well established that isoprene-derived organosulfates are primarily formed via photochemical oxidation mechanisms, involving reactive intermediates such as ISOPOOH and IEPOX under low-NO conditions, often leading to enhanced production during the daytime.

However, in our study, we observed consistently higher CHOS signal intensities at night across all seasons. To address this apparent contradiction, we have revised the discussion to more clearly distinguish between formation processes and partitioning behavior in line 483ff as follows:

"It is well established that isoprene-derived organosulfates are primarily formed via photochemical oxidation mechanisms, involving reactive intermediates such as ISOPOOH and IEPOX under low-NO conditions, often leading to enhanced production during the daytime (Surratt et al., 2007; Surratt et al., 2008). However, in our study, we observed consistently higher CHOS signal intensities at night across all seasons. While OS formation is photochemically driven, the nocturnal enhancement in signal intensities is likely not indicative of in situ nighttime production, but instead reflects more efficient partitioning into the particle phase during cooler nighttime conditions. This explanation is consistent with the findings of Gómez-González et al. (2012) and Kourtchev et al. (2014b), who also reported strong diurnal differences driven by temperature-dependent gas-particle partitioning. "

Lines 416 and 430: Please specify in the appendix the number of standard compounds used and which standard compounds were used, and indicate which substances each standard identified or confirmed.

We thank the reviewer for the helpful suggestion. As requested, we have included a specification of the standard compounds used in the analysis in Table S5 in the supplementary information of the revised manuscript. Since LC was used upstream the ESI-HR-Orbitrap-MS

system, compounds could be assigned unambiguously by comparing the exact mass and retention time of the compound to be identified and the authentic standard used.

Lines 598–600: Previous studies have shown that organic sulfur emissions from soil increase with soil moisture, suggesting that forest soil sources may be stronger during the rainy season. Therefore, it is recommended that the authors further explore whether the vertical distribution of CHOS and CHONS is influenced by soil emissions—especially during nighttime or stable boundary layer conditions. A decreasing concentration gradient from bottom to top could support this hypothesis and strengthen the discussion on the formation and source attribution of CHOS/CHONS compounds.

We appreciate the reviewer's suggestion to further explore the potential influence of forest soil emissions on the vertical distribution of CHOS and CHONS compounds. Indeed, previous studies have highlighted the role of soil-derived organic sulfur emissions under moist conditions. However, the scope of this study is to provide a comprehensive, seasonally averaged overview of the organic aerosol (OA) composition at the ATTO site. As shown in Table I (background OA characteristics) the wet season 2018 revealed 11-18 % of CHOS compounds whereas the contribution for the wet season 2019 is only 2-4 % and for the dry season 2018 4-5 %. The dry season 2019 had almost no significant contribution of CHOS compounds (0-1 %). CHONS compounds showed more or less the same contribution for all seasons (0-2 %). For the total OA characteristics (background+variable compounds), the wet seasons 2018 and 2019 showed the lowest contribution of CHOS compounds (9-11 % and 17-19 % ) whereas the dry seasons 2018 and 2019 exhibited values from 22-23 % and 17-19 % respectively. Therefore, the effects of enhanced soil emissions should be studied event-based and may be masked by temporal averaging. Multiple sources and formation pathways contribute to CHOS/CHONS compounds. For instance, anthropogenic emissions and biomass burning, which are more prevalent during the dry season, lead to elevated CHOS/CHONS levels and complicate a source-specific interpretation based on seasonally averaged data. Therefore, a more targeted approach focusing on specific marker compounds and stratified by meteorological parameters such as rainfall or boundary layer stability would be more appropriate to assess soil-driven emissions.

In fact, such a targeted dataset with rain-event-resolved sampling and focus on selected CHOS compounds has already been collected and will be the subject of a dedicated follow-up publication. This future work will aim to directly address the role of forest soil emissions under varying hydrometeorological conditions and their vertical distribution.

**Citation**: https://doi.org/10.5194/egusphere-2025-141-RC2

Author comment:

We hope that the revised manuscript and the responses provided here fully address the reviewer's concerns. We would be happy to clarify any remaining points.